# FISHER INFORMATION FOR ROBUST FEDERATED CROSS-VALIDATION

## ABSTRACT

When training data are fragmented across batches or federated-learned across different geographic locations, trained models manifest performance degradation. That degradation partly owes to covariate shift induced by data having been fragmented across time and space and producing dissimilar empirical training distributions. Each fragment's distribution is slightly different to a hypothetical unfragmented training distribution of covariates, and to the single validation distribution. To address this problem, we propose Fisher Information for Robust fEderated validation (**FIRE**). This method accumulates fragmentation-induced covariate shift divergences from the global training distribution via an approximate Fisher information. That term, which we prove to be a more computationally-tractable estimate, is then used as a per-fragment loss penalty, enabling scalable distribution alignment. FIRE outperforms importance weighting benchmarks by $5.1\%$ at maximum and federated learning (FL) benchmarks by up to $5.3\%$ on shifted validation sets.

## 1 INTRODUCTION

Machine learning models and systems demonstrate strong predictive performance for cross-sectional data as long as the test data distribution aligns with the training distribution. However, these systems often fail to generalize when the test distribution diverges from training distribution Hendrycks & Dietterich (2019a); Taori et al. (2020). These learning-based systems play a critical role in real-world decision-making scenarios. For instance, a pneumonia detection model trained on chest X-rays from specific hospitals may perform poorly when deployed in new geographic regions due to covariate shift (features training distribution $P_{tr}(x)$ differs from test distribution $P_{tst}(x)$) Gardner et al. (2023). Similarly, fraud detection systems may struggle to adapt across regions where fraud patterns vary significantly Guan et al. (2024). Such failure occurs because standard cross-validation assumes that data is independently and identically distributed Moreno-Torres et al. (2012). The assumption does not hold in real-world as data-sets evolve increasingly *fragmented across time, location or devices*, a phenomenon which is referred as **fragmentation induced covariate shift (FICS)** Khan et al. (2025a).

While covariate shift is a well-studied problem in machine learning Sugiyama et al. (2007a), however, it is still under-investigated in regimes where it is induced by fragmentation Moreno-Torres et al. (2012); Sugiyama et al. (2007a); Khan et al. (2025a;b). Importance weighting-based methods such as importance weighted cross validation (IWCV) Sugiyama et al. (2007a), density ratio estimation (uLSIF) Kanamori et al. (2012), direct importance estimation Sugiyama et al. (2007b), dynamic importance weighting (DIW) Fang et al. (2020), and generalized importance weighting (GIW) Fang et al. (2023) where machine learning models are trained by assigning weights $w(x)$ to each training example $w(x) = \frac{p_{tst}(x)}{p_{trn}(x)}$ assumes a single source distribution and fine-tuned methods uses feature alignment technique Ganin & Lempitsky (2015), while federated learning approaches ignore local shift during validation process Kairouz et al. (2021). To our knowledge, there are no methods which consider covariate shift caused by fragmentation where batches diverge from the validation set and among each other as well.

Non-IID data in FL is well-studied McMahan et al. (2017); Lu et al. (2024). Classical methods (FedAvg, FedProx, SCAFFOLD, FedDyn) focus on stabilizing convergence under heterogeneity, while recent work (e.g., MOON Li et al. (2021a)) aligns local and global representations to mitigate

client drift. In contrast, FICS targets robustness to a distinct validation distribution ($P_{val}$), a challenge overlooked by prior approaches.

Domain generalization (DG) methods such as FISHR Rame et al. (2022) also leverage the Fisher Information Matrix (FIM), but with a different goal, enforcing invariance across multiple source domains to generalize to unseen targets. In contrast, FIRE aligns models trained on fragmented batches or clients with a fixed validation distribution ($P_{val}$). Specifically, we penalize parameter sensitivities via Fisher penalty in the direction of $P_{val}$, ensuring validation consistent adaptation. Unlike FISHR, which matches FIMs *across* domains, FIRE compares them *against* validation, making it the first method to explicitly handle fragmentation in both batch/fold processing and federated learning.

Recent FL methods address diverse forms of heterogeneity, like LFDKim & Shin (2023) regularizes update directions to reduce client drift, FEDASYang et al. (2024) aligns parameters to handle intra/inter-client inconsistency, and FEDCFA Jiang et al. (2025) mitigates Simpson's paradox via counterfactual samples. While effective, these approaches optimize within-training consistency and overlook robustness to a fixed validation distribution. FIRE instead introduces a Fisher-based regularizer that explicitly aligns each client or batch/fold with $P_{val}$, addressing fragmentation-induced covariate shift.

In summary, FIRE offers a new perspective on distribution shift by leveraging Fisher information for validation alignment under fragmentation. To our knowledge, it is the first unified framework that addresses fragmented batches and federated clients, enabling scalable mitigation of covariate shift relative to a fixed validation distribution. The working mechanism of FIRE is given in following Figure 1.

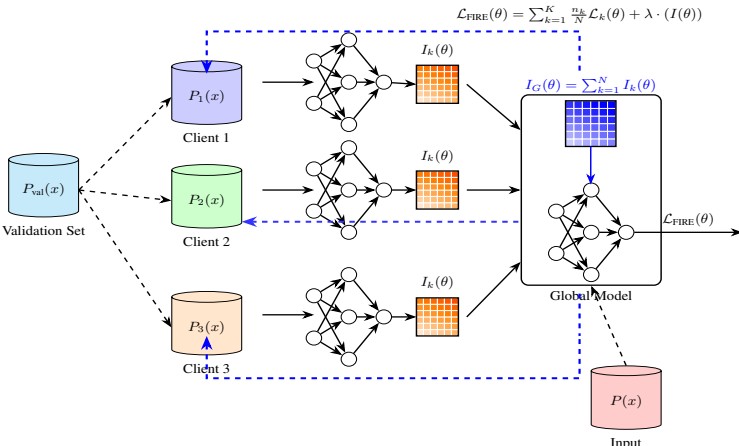

Figure 1: **FIRE** working mechanism in FL setting. The server broadcasts the global model $\theta$ and global FIM $I_G(\theta)$ to all clients. Each client $k$ computes its local FIM $I_k(\theta)$ using the shared validation set $P_{val}(x)$. Clients perform a local update regularized by $I_G(\theta)$ and send their local FIMs back to the server. The server then aggregates the client FIMs (e.g., $I_G(\theta) = \sum_{k=1}^{N} \frac{n_k}{N} I_k(\theta)$) to update the global FIM for the next round. This unified approach ensures model alignment with the target validation distribution in both settings.

## CONTRIBUTIONS

1. We formally define fragmentation-induced covariate shift (FICS) in federated and batch/fold setting, showing the performance of existing methods is compromised by a non-iid (non-independent and identically distributed) cross-validation split.

2. We propose FIRE, the first method to remediate FICS via Fisher information to tractably estimate a function of network parameters. Our method allows memory cost to be linear in the size of a dataset fragment.

3. We extend FIRE to federated learning with minimum communication overhead, outperforming FL baselines.

---

**Algorithm 1** FIRE: Batchwise Fisher accumulation for covariate-shift remediation

---

**Require:** Batches $\{B_i\}_{i=1}^m$, validation set $V$, learning rate $\eta$, penalty $\lambda$, momentum $\alpha$, mixing weight $\mu \in [0, 1]$
**Ensure:** Robust model parameters $\theta$
  1: Initialize $\theta_0$, $I_G \leftarrow \mathbf{0}$
  2: **Precompute (or periodically update) validation FIM:**
  3: $I_V(\theta) \leftarrow \mathbb{E}_{(x,y) \sim V} \left[ \nabla_\theta \log p(y|x; \theta) \nabla_\theta \log p(y|x; \theta)^\top \right]$
  4: **for** each batch $B_i \in \{B_1, \ldots, B_m\}$ **do**
  5:    Compute batch FIM (mini-batch estimate):
  6:    $I_{B_i}(\theta) \leftarrow \mathbb{E}_{(x,y) \sim B_i} \left[ \nabla_\theta \log p(y|x; \theta) \nabla_\theta \log p(y|x; \theta)^\top \right]$
  7:    Form combined per-batch FIM (mix validation and batch):
  8:    $I_i(\theta) \leftarrow \mu\, I_{B_i}(\theta) + (1 - \mu)\, I_V(\theta)$
  9:    Update global FIM with momentum:
 10:    $I_G(\theta) \leftarrow \alpha I_G(\theta) + (1 - \alpha) I_i(\theta)$
 11:    Perform SGD update with FIM-based regularization:
 12:    $\theta \leftarrow \theta - \eta \Big( \nabla_\theta \mathcal{L}(B_i) + \lambda\, I_G(\theta)\, \nabla_\theta \mathcal{L}(B_i) \Big)$
 13: **end for**
 14: **return** $\theta$

---

## 2 METHOD DEVELOPMENT

### 2.1 NOTATIONAL SETUP

In FL context, lets assume that there are $K$ number of clients having their own datasets like: $\mathcal{D}_k = \{(x_i^k, y_i^k)\}_{i=1}^{n_k}$, where $x_i^k$ are the covariates, $y_i^k$ are the labels, while $n_k$ are the number of samples for client $k$. The data distribution $P_k(x, y)$ may differ across clients, leading to covariate shift (i.e., $P_k(x) \neq P_{k'}(x)$ for clients $k$ and $k'$).

### 2.2 UNIFIED FRAMEWORK FOR FRAGMENTED AND FEDERATED LEARNING

Our method, FIRE, handles both traditional fragmented data settings (batches/folds) and federated learning (FL) scenarios under a unified framework. While these two settings differ in data partitioning, they share the core challenge of covariate shift between training and validation distributions. We formalize this connection below:

- In fragmented setting, data split into $k$ batches $\{B_i\}_{i=1}^m$ with distributions $P_i(x) \neq P_{val}(x)$. Our goal is to minimize validation loss $\mathcal{L}_\theta = \mathbb{E}_{(x,y) \sim P_{val}}[\ell(y, f_\theta(x))]$ by aligning $P_i$ with $P_{val}$ via FIM.

- In FL settings the data is partitioned across $K$ clients $\{D_k\}_{k=1}^K$ with $P_k(x) \neq P_{val}(x)$. Goal is to minimize the global loss $\mathcal{L}_{(\theta)} = \sum_{k=1}^K \frac{n_k}{N} \mathcal{L}_k(\theta)$ while ensuring client models generalize to $P_{val}$.

The common idea in both settings is estimating and mitigating covariate shift. Fragmented settings computes FIM $I_i(\theta)$ for batches/folds while FL computes it for clients $I_k(\theta)$ to measure $\mathcal{D}_{KL}(P_i || P_{val})$. During the remediation phase, both settings penalize the loss with $\lambda.I(\theta)$ where $I(\theta)$ is (batchwise or clientwise) aggregated FIM. The FIRE algorithm 1 applies identically in both cases. For batches, $I_G(\theta)$ accumulates shift across sequential batches while in FL scenario, $I_G(\theta)$ is the weighted average of client FIMs. We assume access to a small public validation set $V$ that is representative of the target distribution. For simplicity of exposition, Figure 1 shows this set being shared with clients. In practice, if sharing raw examples is undesirable, the server can compute the validation FIM once (or periodically) and broadcast a compressed approximation (e.g., diagonal) to clients, thereby preserving privacy.

## 2.3 Federated learning specifics

Now we detail our method FIRE practical considerations communication efficiency, scalability, and comparisons to standard FL baselines ensuring feasibility in real-world deployments.

**Communication efficiency.** FIRE transmits client FIM to the server once per global round. For a model with $d$ parameters, each client sends $\mathcal{O}(d^2)$ FIM entries (symmetric, so approximately $d^2/2$ values). Following Rothchild et al. (2020), we use a rank-$k$ approximation ($k \ll d$) to reduce the overhead to $\mathcal{O}(kd)$. FIRE adds minimal overhead compared to gradient transmission ($\mathcal{O}(d)$ per client), as FIMs are aggregated infrequently (every 5 rounds).

**Scalability.** The FIRE framework per-client computation scales as: FIM Cost $= O(b \cdot d^2)$, where b is the batch size. For large d, we approximate $I_k(\theta)$ as diagnol, reducing cost to $O(b.d)$ Kingma et al. (2020).

## 2.4 Problem formulation

Let $\mathcal{D} = \{B_i\}_{i=1}^m$ be a fragmented dataset into $m$ number of batches with batch distribution $P_i(x) \neq P_{val}(x)$, $P_i(x)$ is an arbitrary batch distribution while $P_{val}(x)$ is validation set distribution. Our goal is to minimize loss $\mathcal{L}_\theta$:

$$\mathcal{L}_\theta = \mathbb{E}_{(x,y) \sim P_{val}}[\ell(y, f_\theta(x))]. \tag{1}$$

A principled metric over the probability space distribution is required for measuring the amount of shift between fragmented batches and validation set Khan et al. (2025a). The KL divergence Kullback & Leibler (1951) is a natural choice for batch comparison and its close connection to cross entropy loss commonly used in neural networks. In practice KL divergence is often used as mean-field approximation, where the posterior $q(\theta)$ assumed to be Gaussian and parametrized by covariance of networks weights. However, access to the Hessian of the loss with respect to model $f(\theta)$ is required for computing this term, which is infeasible in high-dimensional settings. To circumvent this, Pascanu & Bengio (2013) proposed approximation of Hessian $\mathbb{E}\left[-\frac{\partial^2 \log p(X|\theta)}{\partial\theta\partial\theta^T}\right]$ using FIM (Fisher information matrix) $I(\theta)$, which is more tractable alternative that also captures the second order information and can be estimated using the expected values and variance of the gradients Nishiyama (2019).

## 2.5 Fisher Information Approximation

Fisher information for each batch $B_i$ and validation set $V$ is computed by:

$$I_i(\theta) = \mathbb{E}_{x \sim V}\left[\nabla_\theta \log p(y|x; \theta)\nabla_\theta \log p(y|x; \theta)^T\right]. \tag{2}$$

where $\mathbb{E}_{x \sim V}$ is expectation over validation set and $\nabla_\theta \log p(y|x; \theta)$ is log-likelihood of model $f(\theta)$ predictions with respect to parameters $\theta$. $I_i(\theta)$ approximates the curvature of $D_{\mathrm{KL}}(P_i \| P_{\mathrm{val}})$.

**assumption 2.1 (Model regularity and bounds)** *The conditional model $p(y \mid x; \theta)$ satisfies the following for all $x, y, \theta, \theta' \in \mathbb{R}^d$:*

    *(R1) (Lipschitz Hessian) $\|\nabla_\theta^2 \log p(y \mid x; \theta') - \nabla_\theta^2 \log p(y \mid x; \theta)\|_{\mathrm{op}} \leq \beta\|\theta' - \theta\|_2$.*

    *(R2) (Bounded score) $\|\nabla_\theta \log p(y \mid x; \theta)\|_2 \leq G$ almost surely; hence the local Fisher satisfies $\|F_x(\theta)\|_{\mathrm{op}} \leq G^2 =: M$.*

    *(R3) (Regularity) For every $x, \theta$ we have $\mathbb{E}_{y \sim p(\cdot|x;\theta)}[\nabla_\theta \log p(y \mid x; \theta)] = 0$.*

**assumption 2.2 (Data proximity)** *Let $P_i(x)$ and $P_{\mathrm{val}}(x)$ be two marginal distributions. Define the Radon–Nikodym derivative $r(x) = \frac{dP_i}{dP_{\mathrm{val}}}(x)$ with assumption $|r(x) - 1| \leq \gamma < 1$ for all $x$.*

**Lemma 2.3 (Marginal KL bound)** *Under Assumption B.2 we have $D_{\mathrm{KL}}(P_i(x)\|P_{\mathrm{val}}(x)) = \mathbb{E}_{x \sim P_{\mathrm{val}}}[r(x) \log r(x)] \leq C_1\gamma^2 + C_1'\gamma^3$, where one can take for instance $C_1 = \frac{1}{2(1-\gamma)}$, $C_1' = \frac{1}{3(1-\gamma)^2}$. see proof detail in appendix B.4*

**Lemma 2.4 (Local conditional KL quadratic expansion)** *Fix $x$. Under Assumption B.1 and for two parameter vectors $\theta_i, \theta_{\text{val}}$ with $\|\theta_i - \theta_{\text{val}}\|_2 \leq \delta$, the conditional KL admits the expansion $D_{\text{KL}}\big(p(\cdot \mid x; \theta_i) \,\big\|\, p(\cdot \mid x; \theta_{\text{val}})\big) = \frac{1}{2}(\theta_i - \theta_{\text{val}})^\top F_x(\theta_i)(\theta_i - \theta_{\text{val}}) + R_x$, with the remainder bounded by $|R_x| \leq \frac{\beta}{6}\delta^3 G$, so in particular $|R_x| \leq \frac{\beta G}{6}\delta^3$ see proof detail in appendix B.5.*

**Theorem 2.5 (KL divergence bound via Fisher information)** *Suppose Assumptions B.1 and B.2 hold. Let $\theta_i, \theta_{\text{val}}$ satisfy $\|\theta_i - \theta_{\text{val}}\|_2 \leq \delta$. Then*

$$D_{\text{KL}}(P_i\|P_{\text{val}}) \leq \tfrac{1}{2}(\theta_i - \theta_{\text{val}})^\top F_{\text{val}}(\theta_{\text{val}})(\theta_i - \theta_{\text{val}}) \\ + C_1\gamma^2 + C_1'\gamma^3 + C_2\,\gamma\,\delta^2 + C_3\,\beta G\,\delta^3, \tag{3}$$

*where $F_{\text{val}}(\theta) = \mathbb{E}_{x \sim P_{\text{val}}}[F_x(\theta)]$, and one may take $C_1 = \frac{1}{2(1-\gamma)}$, $C_1' = \frac{1}{3(1-\gamma)^2}$, $C_2 = \frac{M}{2}$, $C_3 = \frac{1}{6}$.*

see proof detail in appendix B.3.

## 2.6 Connection to Federated Learning

Fragmentation induced covariate shift occurs when sequence of batches (i.e clients in federated leaning) affects the covariates distributions. In federated learning (FL) the data arrives similarly in non-iid manner across clients where the clients are often at different geographic locations. The data distributions differs due to temporal, geographic or user-specific factors which leads to covariate shift McMahan et al. (2017); Du et al. (2022); Ramezani-Kebrya et al. (2023). In FL setting, the model trained for one client data may not be able to generalize to other client data. Prior FL methods such as FedProx and SCAFFOLD Karimireddy et al. (2020) try to mitigate client shift via regularization or variance reduction, they ignore validation-time shift—the misalignment between a client's local data and the global validation set. In our method we address this problem by Fisher information penalty which can be applied in FL setting as well for model generalization. As in batchwise setting, the knowledge about data density of training batch is accumulated to penalize the loss in subsequence batches. In FL as the data is distributed across clients, the same approach can be applied here by accumulating knowledge about data density of each client can be used by the global model for correction of covariate shift across clients. The goal of FIRE in FL setting where data is distributed across clients, is to adapt to covariate shift while training the global model.

$$\mathcal{L}(\theta) = \sum_{k=1}^{K} \frac{n_k}{N} \mathcal{L}_k(\theta) \tag{4}$$

where $\mathcal{L}_k(\theta)$ is loss of local client and $N = \sum_{k=1}^{K} n_k$ is number of samples across all clients.

In FL, the global model leads to poor generalization due to covariate shift arises by the change in features distributions $p(x)$ across clients. FIRE corrects this covariate shift by penalizing the loss function with Fisher information penalty like batch/fold setting. In FL setting, FIRE computes the FIM $I_K(\theta)$ for each client $k$ distribution i.e $I_k(\theta) = \mathbb{E}_{(x,y) \sim P_k(x,y)}\left[-\frac{\partial^2 \log p(y|x;\theta)}{\partial\theta\partial\theta^2}\right]$. Th term $I_k(\theta)$ provides information about local client $k$ data distribution and captures the curvature of local loss function $\mathcal{L}_k(\theta)$. This term is then integrated into local client loss as penalty term such as:

$$\mathcal{L}_k(\theta) = \mathcal{L}_k(\theta) + \lambda \cdot (I_k(\theta)) \tag{5}$$

After computing FIM for all clients, global model then find weight average of these local FIMs as:

$$I(\theta) = \sum_{k=1}^{K} \frac{n_k}{N} I_k(\theta) \tag{6}$$

The global FIM contain curvature of all local clients loss. In FL, this term is integrated into global model loss function as penalty term to correct for covariate shift i.e:

$$\mathcal{L}_{\text{FIRE}}(\theta) = \sum_{k=1}^{K} \frac{n_k}{N} \mathcal{L}_k(\theta) + \lambda \cdot (I(\theta)) \tag{7}$$

where $\lambda$ is hyper-parameter for controlling strength of penalty. In optimization phase, the clients computes its local FIM $I_k(\theta)$ and local gradient $\nabla \mathcal{L}_k(\theta)$ first then the global model aggregates these FIMs and local gradients for the global update.

$$\theta \leftarrow \theta - \eta \left( \sum_{k=1}^{K} \frac{n_k}{N} \nabla L_k(\theta) + \lambda \cdot \nabla(I(\theta)) \right) \tag{8}$$

where $\eta$ is global model learning hyper-parameter. The introduced Fisher information acts as regularizer which penalizes the model parameters which leads to covariate shift. It helps the global model in better generalization across the clients with different data distributions. The integration of this penalty into global model make it robust to covariate shift, which is common challenge in FL settings due to non-iid nature of client data. The results show in Table 5 provide empirical evidence to the effectiveness of this method.

## 3 RELATED WORK

**Covariate shift.** In supervised machine learning the model expects that sample test distribution follows same training distribution Vapnik & Vapnik (1998); Schölkopf & Smola (2002); Duda et al. (2006). However, this assumption does not hold in real-world due to non-stationary environment or samples bias selection Quiñonero-Candela et al. (2022); Sugiyama & Kawanabe (2012). The term *Covariate shift* was coined by Shimodaira Shimodaira (2000) where covariates (features) training distribution differs from test distribution. Covariate shift is common in many real-world applications, such as emotion recognition Jirayucharoensak et al. (2014) speaker identification Yamada et al. (2010) and brain-computer interface Li et al. (2010).

**Importance weighting.** Importance weighting (IW) is the most common approach used for adaptation under distribution shift. IW estimates density ratio between training and test distribution and uses it for reweighting the training loss during optimization Shimodaira (2000); Sugiyama et al. (2007b). Kernel mean matching (KMM) Gretton et al. (2009) minimize maximum mean discrepancy (MMD) in kernel Hilbert space to align training and test distribution. Sugiyama et al. (2007b) uses direct density ratio estimation KLIEP for covariate shift adaptation. Recent methods like dynamic importance weighting (DIW) Fang et al. (2020) address distribution shift by updating the importance weights during stochastic optimization while avoiding the offline density ratio estimation. Generalize importance weighting (GIW) Fang et al. (2023) adapt to distribution shift by dynamically estimating the importance weights for training examples using gradient-based optimization without explicit density ratio estimation.

**Domain Adaptation.** In domain adaptation, distribution shift is addressed by aligning the source and target domains with assumption that target data is accessible during training. Adversarial methods such as DANN Ganin et al. (2016) learn domain-invariant features using gradient reversal, while discrepancy-based methods minimize divergence metrics such as MMD Tzeng et al. (2014) or CORAL Sun & Saenko (2016). Recent work, such as Zhao et al. (2018), extends domain adaptation to multi-source settings or partial adaptation Cao et al. (2022). However, these methods fail when target data is inaccessible (e.g. fragmented batches) or when distribution shift occurs across clients in federated learning (FL) settings.

**Federated learning under covariate shift.** Non-IID client data induces covariate shift in FL, challenging global model training. FedAvg McMahan et al. (2017) averages local updates but struggles with client-specific shifts; extensions such as FedBN Li et al. (2021b), FedProx Li et al. (2020), clustering Zhang et al. (2020a), and meta-learning Jiang et al. (2019) improve stability under heterogeneity. Contrastive methods like MOON Li et al. (2021a) reduce client drift, while recent advances (LfD Kim & Shin (2023), FedAS Yang et al. (2024), FedCFA Jiang et al. (2025)) address drift, inconsistency, and aggregation bias. Yet, these approaches focus on training- or aggregation-time robustness and overlook distribution shift at *validation* time. Our method closes this gap by introducing Fisher-driven alignment with a fixed validation distribution.

## 4 EXPERIMENTS

We evaluated the effectiveness of our method **FIRE** in fragmented (batches) and in federated settings (clients) on standard benchmarks.

**Datasets.** FIRE performance evaluated on 39 total datasets. The datasets includes: F-MNIST Xiao et al. (2017), K-MNIST Clanuwat et al. (2018) and MNIST-C Mu & Gilmer (2019) which is also used by Sugiyama et. al, as benchmark for covariate shift adaptation . For inducing shift in these three datasets, we follow the same procedure given in One-step approach Zhang et al. (2020b)[1] Covariate shift is induced on five tabular datasets—Australian, Breast Cancer, Diabetes, Heart, and Sonar from the KEEL repository Alcalá-Fdez et al. (2011), following the procedure of Sugiyama et al. Zhang et al. (2020b), originally adapted from Cortes et al. Cortes et al. (2008). All other datasets, including image datasets such as MNIST LeCun et al. (2010), EMNIST Cohen et al. (2017), QMNIST Yadav & Bottou (2019), Kannada-MNIST Prabhu (2019), CIFAR-10 and CIFAR-100 Coates et al. (2011), SVHN Netzer et al. (2011), Caltech101 Fei-Fei et al. (2004), Tiny-ImageNet Krizhevsky et al. (2009), STL-10 Krizhevsky et al. (2009), P-MNIST Mu & Gilmer (2019), and corruption variants CIFAR-10-C and CIFAR-100-C Hendrycks & Dietterich (2019b), as well as 27 additional binary classification datasets from KEEL Alcalá-Fdez et al. (2011), are used under their standard published settings for evaluating FIRE performance.

**Model architecture.** In fragmented (batches/folds) setting, we use a five-layer convolutional neural network (CNN) with softmax cross-entropy loss for all image-based benchmarks. The architecture consist of two convolutional layers with pooling followed by three fully connected layers. Hyperparameters (optimizer = Adam, activation = softmax, and training epochs = 100) are fixed across image datasets. For tabular datasets, we employ a consistent multi-layer perceptron (MLP) architecture with a single hidden layer of 4 neurons. The tabular setting uses hyperparameters (activation = ReLU, optimizer = Adam, and epochs = 1500). All reported accuracies are averaged over 100 independent runs.

In federated learning setting we used a three-layer fully connected neural network with ReLU activations, mapping 784-dimensional inputs to 10 output classes via hidden layers of size 512 and 256. This architecture is used consistently across all clients. The model is trained with FIM penalty to mitigate covariate shift in FL setting.

The penalty coefficient $\lambda$ is held constant across all datasets, calibrated using a batch/fold configuration as shown in Figure 2.

All baselines are implemented in TensorFlow 2.11, and code is available at the anonymous hyper-link[2]. We reproduce baseline results as reported in the original publications More implementation details can be found in appendix C.1.

**Evaluation metrics.** For performance evaluation of our method FIRE we used accuracy as a metric. The accuracy metric remains consistent across all our experiments like in fragmented setting (image/tabular), and in FL setting too.

**Experimental design.** We evaluate FIRE on fragmented data (batches/folds) and in FL settings (clients) using five sets of experiments, with standard cross-validation (st-CV) as the baseline. The experiments are:

- **Covariate shift in batch settings:** We evaluate st-CV on both the integral dataset and its fragmented (batched) versions to examine the impact of fragmentation-induced covariate shift. The experiments are conducted on 13 image-based and. Results are reported in Table 6.

- **FIRE shift mitigation in batch settings:** We apply FIRE with FIM-based penalty on both integral and fragmented batches to evaluate its effectiveness in mitigating covariate shift. The results are presented in Table 7.

- **Covariate shift in fold settings:** To assess the impact of fragmentation-induced covariate shift in fold settings, we conduct experiments on 26 tabular datasets. Results are presented in Appendix Table 9.

- **FIRE shift mitigation in fold settings:** We assess FIRE's efficacy in mitigating covariate shift in fold settings using tabular data Results can found in Table 9.

---

[1]Each training image $I_i$ is rotated by angle $\theta_i$, with $\theta_i/180°$ drawn from distribution Beta$(a, b)$. For test images $J_i$, the rotation angle $\phi_i$ is drawn from Beta$(b, a) = (2, 4), (2, 5)$, and $(2, 6)$.
[2]FIRE

- **Comparison of FIRE with FL state-of-the-art:** We evaluate FIRE's robustness against state-of-the-art (SOTA) methods like FedAvg McMahan et al. (2017), SCAFFOLD Karim-ireddy et al. (2020), MOON Li et al. (2021a), LfD Kim & Shin (2023), FedAS Yang et al. (2024), and FedCFA Jiang et al. (2025). Results are presented in Table 5.

## 5 RESULTS AND DISCUSSION

### 5.1 COVARIATE SHIFT AND BATCHES/FOLDS SETTINGS

Table 6 shows that dataset fragmentation into batches consistently degrades both average and batch-wise accuracy compared to the st-CV baseline, due to the covariate shift it induces. On image-based datasets the fragmentation leads to over 36% and 60% drop in average accuracy across 2, 10, and 20 batches, indicating an effect from induced shift.

It can be noticed from table that effect of fragmentation frequency also effects accuracy. Figure in appendix 3 and Table 6 show that accuracy loss increases with fragmentation frequency such as 52.1% for 20 batches, 43.7% for 10, and 36.3% for 2. Fewer batches offer greater data support, partially mitigating the shift.

st-CV results under varying fold settings are reported in Table 8. In fold settings the baseline remains same (i.e st-CV) to test whether data fragmentation induces distribution shift. As shown in Table 8, the accuracy consistently degrades with increasing folds, indicating shift. We report mean accuracy for each fold setting; $\mu_3$, $\mu_4$, and $\mu_5$ denote averages for (2, 5, and 10) folds, respectively.

### 5.2 FIRE SHIFT MITIGATION IN BATCH/FOLDS SETTINGS

**FIRE mitigation in batch settings.** FIRE, effectively mitigates FICS in no-shift settings. As shown in Table 7, column $\Delta_3$, it improves average accuracy by over 10% across batch fragmentation levels (20, 10, 2), consistently across datasets. This gain stems from FIRE ability to extract and retain batch-sequence information while regularizing the model. Cross-batch comparisons show significant improvements in remediated shifts when aligned at the same sequence index. For example, under 20-way fragmentation on F-MNIST, batch $B_n$ improves from 70.6% (with shift, Table 6) to 81.9% (remediated, Table 7), a 16% increase. These results highlight FIRE robustness across batch positions and fragmentation settings.

**FIRE mitigation in fold settings.** Table 8 presents st-CV results under $k$-fold settings, where st-CV serves as the baseline to test our hypothesis that data the fragmentation induces distribution shift. As shown in Tables 9 and 8, accuracy degrades with increasing fold count, confirming that finer fragmentation amplifies shift. We report mean accuracies for each setting: $\mu_3$, $\mu_4$, and $\mu_5$ correspond to (2, 5, and 10) folds, respectively.

Tables 9 and 8 present $k = 2, 5, 10$ fold results for tabular datasets. In Table 9, accuracy gaps $\Delta_5 = \mu_3 - \mu_6$, $\Delta_6 = \mu_4 - \mu_7$, $\Delta_7 = \mu_5 - \mu_8$ highlight the impact of induced shift. Our method improves accuracy by up to 28.3% across all settings.

Table 1: FIRE benchmarking with SOTA on image datasets ($\Delta_1$ % = FIRE - SOTA)

| Dataset | Shift Level (a, b) | ERM | EIWERM | One-step | FIRE | $\Delta_1$% |
|---|---|---|---|---|---|---|
| F-MNIST | (2, 4) | 64.6 ± 0.17 | 71.3 ± 0.06 | 74.5 ± 0.08 | 78.3 ± 0.14 | ↑ 5.10% |
| | (2, 5) | 54.5 ± 0.54 | 57.9 ± 0.29 | 55.6 ± 0.20 | 57.2 ± 0.31 | ↑ 2.87% |
| | (2, 6) | 36.3 ± 0.34 | 42.5 ± 0.55 | 44.8 ± 0.25 | 45.4 ± 0.20 | ↑ 1.33% |
| K-MNIST | (2, 4) | 67.1 ± 0.18 | 69.7 ± 0.24 | 68.8 ± 0.12 | 72.2 ± 0.44 | ↑ 4.94% |
| | (2, 5) | 55.0 ± 0.26 | 52.2 ± 0.19 | 59.5 ± 0.16 | 61.6 ± 0.09 | ↑ 3.52% |
| | (2, 6) | 39.2 ± 0.30 | 38.4 ± 0.93 | 43.1 ± 0.55 | 43.8 ± 0.35 | ↑ 1.62% |
| MNIST-C | (2, 4) | 63.6 ± 0.91 | 80.5 ± 0.08 | 85.2 ± 0.17 | 86.3 ± 0.28 | ↑ 1.29% |
| | (2, 5) | 43.8 ± 0.18 | 60.4 ± 0.47 | 78.4 ± 0.32 | 79.3 ± 0.67 | ↑ 1.14% |
| | (2, 6) | 33.3 ± 0.49 | 53.8 ± 0.13 | 64.2 ± 0.67 | 64.9 ± 0.37 | ↑ 1.09% |

### 5.3 COMPARISON WITH STATE-OF-THE-ART

**FIRE in comparison to importance weighting methods.** Tables 1 ,2, 5 present stat-of-the result comparison of FIRE. It is shown that FIRE consistently outperforms the existing methods, including

Table 2: Benchmarking with SOTA on tabular datasets. Mean accuracy with *Wilcoxon signed-rank test* Wilcoxon (1992) at significance level 5% across various datasets with induced covariate shift ($\Delta_2$ % = FIRE - SOTA)

| Dataset | ERM | uLSIF | RuLSIF | One-step | FIRE | $\Delta_2$% |
|---|---|---|---|---|---|---|
| heart | $65.3 \pm 9.91$ | $64.1 \pm 11.4$ | $63.2 \pm 11.7$ | $74.3 \pm 10.9$ | $\mathbf{78.1 \pm 5.90}$ | ↑ 5.11% |
| sonar | $61.9 \pm 12.9$ | $64.6 \pm 13.2$ | $63.7 \pm 13.5$ | $67.6 \pm 12.4$ | $\mathbf{70.4 \pm 5.91}$ | ↑ 4.14% |
| diabetes | $54.2 \pm 8.88$ | $57.5 \pm 7.66$ | $55.7 \pm 8.63$ | $62.9 \pm 6.36$ | $\mathbf{64.3 \pm 12.6}$ | ↑ 2.22% |
| australian | $67.9 \pm 16.8$ | $69.3 \pm 16.3$ | $69.6 \pm 15.1$ | $74.4 \pm 12.7$ | $\mathbf{75.7 \pm 5.86}$ | ↑ 1.74% |
| breast cancer | $78.3 \pm 13.4$ | $79.9 \pm 12.4$ | $78.6 \pm 12.9$ | $77.4 \pm 10.1$ | $73.6 \pm 4.62$ | ↓ 4.90% |

EIWERM, RuLSIF, and One-step. In Table 1 ($\Delta_1$), FIRE achieves up to 5.10% improvement over One-step across high-dimensional datasets under various shift levels. This performance gain likely stems from its ability to retain prior knowledge, unlike EIWERM, which may suffer from over-flattened importance weights, and ERM, which struggles under distribution shifts.

Similarly, Table 2 ($\Delta_2$) shows up to 1.74% improvement at minimum and 5.11% at maximum on 4 out of 5 datasets. Where other methods like, uLSIF and RuLSIF underperform, possibly due to sensitivity to edge examples. Appendix Table 9 report results for $k \in \{2, 5, 10\}$ folds. Notably, FIRE achieves up to 28.3% improvement in average accuracy across all k-fold settings ($\Delta_5 = \mu_3 - \mu_6$, $\Delta_6 = \mu_4 - \mu_7$, $\Delta_7 = \mu_5 - \mu_8$).

**FIRE Outperforms State-of-the-Art in Federated Learning under Non-IID Shift.** Results in Table 5 demonstrate the effectiveness of FIRE against a comprehensive suite of modern federated learning algorithms. FIRE consistently achieves the highest accuracy across all evaluated datasets, providing a clear and reliable improvement over all other methods.

The $\Delta$ column shows that FIRE delivers a consistent performance gain of 2.8-3.2% over the FL baselines (FedCFA). Notably, the improvement is strongest on the less complex FEMNIST and CIFAR-10 datasets, with gains of 5.3% and 3.0% respectively. On the more challenging CIFAR-100 benchmark, FIRE still achieves a solid 2.8% improvement. This pattern suggests that FIRE's regularization is highly effective, and its relative benefit remains significant even as task difficulty increases. Our proposed method FIRE outperform all baselines including SCAFFOLD, MOON, Fishr, LfD, FedAS and FedAvg.

In addition to accuracy, FIRE exhibits consistently lower standard deviations than all competing methods, underscoring its stability and robustness key properties for practical deployment. These results confirm that explicitly mitigating fragmentation-induced covariate shift via Fisher information offers a complementary advantage, yielding models that are both more generalizable and reliable.

Table 3: Performance on Federated datasets with Non-IID Data. $\Delta$ shows the percentage improvement of FIRE over the best baseline.

| Dataset | FedAvg | SCAFFOLD | MOON | Fishr | LfD | FedAS | FedCFA | FIRE | $\Delta_3$ (%) |
|---|---|---|---|---|---|---|---|---|---|
| FEMNIST | 58.2 | 63.8 | 64.3 | 63.9 | 64.7 | 64.9 | 65.1 | **68.6** | ↑5.3 |
| | (3.1) | (2.1) | (1.9) | (2.0) | (1.8) | (1.7) | (1.7) | (1.5) | |
| CIFAR-10 | 42.7 | 48.2 | 49.8 | 49.2 | 50.1 | 50.3 | 50.6 | **52.1** | ↑3.0 |
| | (4.5) | (3.0) | (2.4) | (2.6) | (2.3) | (2.2) | (2.1) | (1.9) | |
| CIFAR-100 | 23.4 | 27.1 | 28.2 | 27.8 | 28.5 | 28.6 | 28.8 | **29.6** | ↑2.8 |
| | (2.8) | (2.0) | (1.7) | (1.8) | (1.6) | (1.6) | (1.6) | (1.5) | |

# 6 CONCLUSION

We propose FIRE, a unified framework for mitigating fragmentation-induced covariate shift (FICS) in both batch/fold and federated learning settings. FIRE leverages Fisher information to accumulate and align distribution shifts across sequential batches or clients, addressing a key limitation in existing methods that assume single-source distributions or overlook validation-time shifts. Our theoretical analysis shows that FIRE bounds the KL divergence through Fisher-based regularization, enabling scalable adaptation without density ratio estimation.

FIRE is evaluated across 39 datasets, and we noticed that FIRE outperforms importance weighting methods by up to 5.1% and federated learning baselines by 5.3% under validation-time shifts.

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

## A  THEORETICAL JUSTIFICATION

## B  CONVERGENCE ANALYSIS

**assumption B.1 (Model regularity and bounds)** *The conditional model $p(y \mid x; \theta)$ satisfies the following for all $x, y, \theta, \theta' \in \mathbb{R}^d$:*

*(R1) (Lipschitz Hessian) $\|\nabla_\theta^2 \log p(y \mid x; \theta') - \nabla_\theta^2 \log p(y \mid x; \theta)\|_{\mathrm{op}} \leq \beta \|\theta' - \theta\|_2$.*

*(R2) (Bounded score) $\|\nabla_\theta \log p(y \mid x; \theta)\|_2 \leq G$ almost surely; hence the local Fisher satisfies $\|F_x(\theta)\|_{\mathrm{op}} \leq G^2 =: M$.*

*(R3) (Regularity) For every $x, \theta$ we have $\mathbb{E}_{y \sim p(\cdot \mid x; \theta)}[\nabla_\theta \log p(y \mid x; \theta)] = 0$.*

**assumption B.2 (Data proximity)** *Let $P_i(x)$ and $P_{\mathrm{val}}(x)$ be two marginal distributions. Define the Radon–Nikodym derivative $r(x) = \frac{dP_i}{dP_{\mathrm{val}}}(x)$. Assume*

$$|r(x) - 1| \leq \gamma < 1 \quad \text{for all } x.$$

**Theorem B.3 (KL divergence bound via Fisher information)** *Suppose Assumptions B.1 and B.2 hold. Let $\theta_i, \theta_{\mathrm{val}}$ satisfy $\|\theta_i - \theta_{\mathrm{val}}\|_2 \leq \delta$. Then*

$$\begin{aligned}
D_{\mathrm{KL}}(P_i \| P_{\mathrm{val}}) \leq \tfrac{1}{2}(\theta_i - \theta_{\mathrm{val}})^\top F_{\mathrm{val}}(\theta_{\mathrm{val}})(\theta_i - \theta_{\mathrm{val}}) \\
+ C_1 \gamma^2 + C_1' \gamma^3 + C_2 \, \gamma \, \delta^2 + C_3 \, \beta G \, \delta^3,
\end{aligned} \tag{9}$$

*where $F_{\mathrm{val}}(\theta) = \mathbb{E}_{x \sim P_{\mathrm{val}}}[F_x(\theta)]$, and one may take*

$$C_1 = \frac{1}{2(1 - \gamma)}, \quad C_1' = \frac{1}{3(1 - \gamma)^2}, \quad C_2 = \frac{M}{2}, \quad C_3 = \frac{1}{6}.$$

**Proof B.1** *Start with the decomposition*

$$D_{\mathrm{KL}}(P_i \| P_{\mathrm{val}}) = D_{\mathrm{KL}}(P_i(x) \| P_{\mathrm{val}}(x)) + \mathbb{E}_{x \sim P_i}\big[D_{\mathrm{KL}}(p(\cdot \mid x; \theta_i) \| p(\cdot \mid x; \theta_{\mathrm{val}}))\big].$$

*Apply Lemma 2.3 to bound the marginal term by $C_1 \gamma^2 + C_1' \gamma^3$.*

*For the conditional term, use Lemma 2.4 and average over $x \sim P_i$:*

$$\mathbb{E}_{x \sim P_i}\big[D_{\mathrm{KL}}(\cdot)\big] = \tfrac{1}{2}(\theta_i - \theta_{\mathrm{val}})^\top \mathbb{E}_{x \sim P_i}[F_x(\theta_i)](\theta_i - \theta_{\mathrm{val}}) + \mathbb{E}_{x \sim P_i}[R_x].$$

*Bound the remainder: $|\mathbb{E}_{x \sim P_i}[R_x]| \leq \frac{\beta G}{6} \delta^3 = C_3 \beta G \delta^3$.*

*Now compare $\mathbb{E}_{x \sim P_i}[F_x(\theta_i)]$ to $F_{\mathrm{val}}(\theta_{\mathrm{val}})$ by adding and subtracting intermediate terms:*

$$\mathbb{E}_{P_i}[F_x(\theta_i)] - F_{\mathrm{val}}(\theta_{\mathrm{val}}) = \underbrace{\mathbb{E}_{P_i}[F_x(\theta_i)] - \mathbb{E}_{P_{\mathrm{val}}}[F_x(\theta_i)]}_{(I)} + \underbrace{\mathbb{E}_{P_{\mathrm{val}}}[F_x(\theta_i)] - \mathbb{E}_{P_{\mathrm{val}}}[F_x(\theta_{\mathrm{val}})]}_{(II)}.$$

*For (I), using $|r(x) - 1| \leq \gamma$ and $\|F_x(\theta_i)\|_{\mathrm{op}} \leq M$, we have*

$$\|(I)\|_{\mathrm{op}} \leq M\gamma.$$

*Hence the quadratic form contribution from (I) is at most $\tfrac{1}{2} M \gamma \|\theta_i - \theta_{\mathrm{val}}\|^2 \leq C_2 \, \gamma \, \delta^2$ with $C_2 = \frac{M}{2}$.*

*For (II), by Lipschitzness of the Hessian (Assumption R1) and bounded gradients (R2) one gets a bound $\|\mathbb{E}_{P_{\mathrm{val}}}[F_x(\theta_i)] - \mathbb{E}_{P_{\mathrm{val}}}[F_x(\theta_{\mathrm{val}})]\|_{\mathrm{op}} \leq L_F \|\theta_i - \theta_{\mathrm{val}}\|$ for some $L_F = O(\beta G)$, hence its effect on the quadratic form is $O(\delta^3)$ and is absorbed into the $C_3 \beta G \delta^3$ term (one can make $L_F$ explicit if desired).*

*Combining these bounds yields the inequality equation 9.*

**Lemma B.4 (Marginal KL bound)** *Under Assumption B.2 we have*

$$D_{\mathrm{KL}}(P_i(x) \| P_{\mathrm{val}}(x)) = \mathbb{E}_{x \sim P_{\mathrm{val}}}\big[r(x) \log r(x)\big] \leq C_1 \gamma^2 + C_1' \gamma^3,$$

*where one can take for instance*

$$C_1 = \frac{1}{2(1 - \gamma)}, \qquad C_1' = \frac{1}{3(1 - \gamma)^2}.$$

**Proof B.2** *Write* $r(x) = 1 + u(x)$ *with* $|u| \leq \gamma$. *Using the Taylor expansion* $\log(1 + u) = u - \frac{u^2}{2} + \frac{u^3}{3(1+\xi)^3}$ *for some* $\xi \in (0, u)$, *we get*

$$r \log r = (1 + u)\Big(u - \tfrac{u^2}{2}\Big) + (1 + u)\frac{u^3}{3(1 + \xi)^3}.$$

*Thus* $r \log r = u + \frac{u^2}{2} + R_m(u)$ *where* $|R_m(u)| \leq \frac{|u|^3}{3(1-\gamma)^2}$. *Integrating against* $P_{\text{val}}$ *and using* $\mathbb{E}_{P_{\text{val}}}[u] = \mathbb{E}_{P_i}[1] - 1 = 0$ *(mass conservation), we obtain*

$$D_{\text{KL}}(P_i(x)\|P_{\text{val}}(x)) \leq \tfrac{1}{2}\mathbb{E}[u^2] + \tfrac{1}{3(1-\gamma)^2}\mathbb{E}[|u|^3] \leq \tfrac{1}{2(1-\gamma)}\gamma^2 + \tfrac{1}{3(1-\gamma)^2}\gamma^3,$$

*where we used* $\mathbb{E}[u^2] \leq \sup|u| \cdot \mathbb{E}[|u|] \leq \gamma \cdot \gamma = (\gamma^2)$ *and minor algebra to obtain the stated constants.*

**Lemma B.5 (Local conditional KL quadratic expansion)** *Fix* $x$. *Under Assumption B.1 and for two parameter vectors* $\theta_i, \theta_{\text{val}}$ *with* $\|\theta_i - \theta_{\text{val}}\|_2 \leq \delta$, *the conditional KL admits the expansion*

$$D_{\text{KL}}\big(p(\cdot \mid x; \theta_i) \,\big\|\, p(\cdot \mid x; \theta_{\text{val}})\big) = \tfrac{1}{2}(\theta_i - \theta_{\text{val}})^\top F_x(\theta_i)(\theta_i - \theta_{\text{val}}) + R_x,$$

*with the remainder bounded by*

$$|R_x| \leq \frac{\beta}{6}\,\delta^3\,G,$$

*so in particular* $|R_x| \leq \frac{\beta G}{6}\delta^3$.

**Proof B.3** *Taylor-expand* $\log p(y \mid x; \cdot)$ *at* $\theta_i$:

$$\log p(y \mid x; \theta_{\text{val}}) = \log p(y \mid x; \theta_i) + (\theta_{\text{val}} - \theta_i)^\top \nabla \log p(y \mid x; \theta_i) + \tfrac{1}{2}(\theta_{\text{val}} - \theta_i)^\top \nabla^2 \log p(y \mid x; \theta_i)(\theta_{\text{val}} - \theta_i) + r_3,$$

*where by (R1) the third-order remainder satisfies* $|r_3| \leq \frac{\beta}{6}\|\theta_{\text{val}} - \theta_i\|^3$. *Taking expectation under* $y \sim p(\cdot \mid x; \theta_i)$, *the linear term vanishes by (R3). The quadratic term yields the Fisher form with* $F_x(\theta_i) = \mathbb{E}_{y \sim p(\cdot \mid x; \theta_i)}[\nabla \log p \, \nabla \log p^\top]$ *and the integrated remainder is bounded by* $\frac{\beta}{6}\delta^3$ *multiplied by a factor at most* $G$ *coming from integrating the score magnitude; hence the stated bound.*

**corollary B.6 (FIRE Surrogate via Fisher Information)** *Under the assumptions of Theorem* **??***, the divergence between any client distribution* $P_i$ *and the validation distribution* $P_{\text{val}}$ *admits the quadratic Fisher approximation*

$$D_{\text{KL}}(P_i \,\|\, P_{\text{val}}) \approx \tfrac{1}{2}(\theta_i - \theta_{\text{val}})^\top F_{\text{val}}(\theta_{\text{val}})(\theta_i - \theta_{\text{val}}),$$

*with controlled remainder* $O(\gamma^2 + \gamma\delta^2 + \beta\delta^3)$. *Hence, the Fisher Information Matrix (FIM) serves as a tractable surrogate for measuring distributional misalignment, which forms the basis of the* FIRE *regularization principle.*

**remark B.7 (Practical Computation of FIM)** *Although the full Fisher Information Matrix can be computationally expensive to evaluate, in practice FIRE does not require its exact form. Several approximations make it tractable:*

1. ***Mini-batch estimation:*** $F_{\text{val}}(\theta)$ *can be approximated from stochastic gradients on small validation batches.*

2. ***Diagonal or block-diagonal structure:*** *Restricting to diagonal or layer-wise block FIMs significantly reduces memory and computation.*

3. ***Low-rank projections:*** *Randomized sketching and Kronecker-factored approximations (K-FAC) yield efficient surrogates while preserving sensitivity to distributional misalignment.*

*Thus, FIRE leverages Fisher information as a theoretically grounded proxy for KL divergence while remaining computationally feasible in large-scale federated or fragmented learning scenarios.*

This section provides a theoretical analysis perspective of the convergence properties of our FIRE algorithm. We begin with standard assumptions and then we provide the convergence theorem.

## B.1 ASSUMPTIONS

We impose the following standard assumptions (smoothness, bounded stochastic gradients, bounded positive FIM and bounded global FIM) on our loss function $\mathcal{L}(\theta)$ the FIM $I(\theta)$ and the stochastic gradients.

**assumption B.8 (Loss-smoothness)** *The loss function $\mathcal{L} : \mathbb{R}^d \to \mathbb{R}$ is $L$-smooth, if there exists a constant $L > 0$ such that for all $\theta, \theta'$,*

$$\|\nabla\mathcal{L}(\theta) - \nabla\mathcal{L}(\theta')\| \leq L\|\theta - \theta'\|.$$

*Equivalently,*

$$\mathcal{L}(\theta') \leq \mathcal{L}(\theta) + \nabla\mathcal{L}(\theta)^\top (\theta' - \theta) + \tfrac{L}{2}\|\theta' - \theta\|^2.$$

**assumption B.9 (Lower boundedness)** *The loss is bounded below: there exists $\mathcal{L}_\star > -\infty$ such that*

$$\mathcal{L}(\theta) \geq \mathcal{L}_\star \quad \text{for all } \theta.$$

**assumption B.10 (Unbiased stochastic gradients with bounded variance)** *At iteration $t$, let $g_t = \nabla_\theta \mathcal{L}(B_t; \theta^{(t)})$ denote the stochastic gradient on a mini-batch $B_t$. Then, conditioned on $\theta^{(t)}$,*

$$\mathbb{E}[g_t \mid \theta^{(t)}] = \nabla\mathcal{L}(\theta^{(t)}), \qquad \mathbb{E}\Big[\|g_t - \nabla\mathcal{L}(\theta^{(t)})\|^2 \mid \theta^{(t)}\Big] \leq \sigma^2,$$

*for some $\sigma^2 > 0$.*

**assumption B.11 (Bounded global FIM preconditioner)** *The global Fisher Information Matrix (FIM) estimator is updated via an exponential moving average*

$$I_G^{(t)} = \alpha I_G^{(t-1)} + (1-\alpha)I_i^{(t)}, \qquad \alpha \in [0, 1).$$

*Each local FIM $I_i^{(t)}$ is symmetric positive semidefinite, and $I_G^{(0)} \succeq 0$. Consequently, $I_G^{(t)}$ is symmetric PSD for all $t$, and its spectral norm is uniformly bounded:*

$$\|I_G^{(t)}\| \leq G, \qquad \text{for some constant } G > 0.$$

We analyze FIRE as preconditioned SGD on $\mathcal{L}$ with the update $\theta^{(t+1)} = \theta^{(t)} - \eta(I + \lambda I_G^{(t)})\, g_t$, where $g_t = \nabla\mathcal{L}(B_t; \theta^{(t)})$.

**Theorem B.12 (Convergence of FIRE as Preconditioned SGD)** *Under the assumptions, for any step-size $\eta \leq \frac{1}{L(1+\lambda G)^2}$, the iterates satisfy*

$$\frac{1}{T}\sum_{t=0}^{T-1} \mathbb{E}\big[\|\nabla\mathcal{L}(\theta^{(t)})\|^2\big] \leq \frac{2\big(\mathcal{L}(\theta^{(0)}) - \mathcal{L}_\star\big)}{\eta T} + \eta\, L\,(1+\lambda G)^2\, \sigma^2.$$

*Equivalently,*

$$\frac{1}{T}\sum_{t=0}^{T-1} \mathbb{E}\big[\|(I + \lambda I_G^{(t)})\nabla\mathcal{L}(\theta^{(t)})\|^2\big] \leq (1+\lambda G)^2 \left\{ \frac{2\big(\mathcal{L}(\theta^{(0)}) - \mathcal{L}_\star\big)}{\eta T} + \eta\, L\,(1+\lambda G)^2\, \sigma^2 \right\}.$$

*Choosing $\eta = \Theta\big(\frac{1}{(1+\lambda G)L\sqrt{T}}\big)$ yields $\min_{0 \leq t < T} \mathbb{E}\|\nabla\mathcal{L}(\theta^{(t)})\|^2 = O(1/\sqrt{T})$.*

**Proof B.4 (Proof sketch)** *By $L$-smoothness of $\mathcal{L}$ and the update $\Delta_t = -\eta(I + \lambda I_G^{(t)})g_t$,*

$$\mathcal{L}(\theta^{(t+1)}) \leq \mathcal{L}(\theta^{(t)}) + \nabla\mathcal{L}(\theta^{(t)})^\top \Delta_t + \frac{L}{2}\|\Delta_t\|^2.$$

*Take conditional expectation given $\theta^{(t)}$ and use $\mathbb{E}[g_t \mid \theta^{(t)}] = \nabla\mathcal{L}(\theta^{(t)})$ to get the descent term $-\eta\|(I + \lambda I_G^{(t)})^{1/2}\nabla\mathcal{L}(\theta^{(t)})\|^2$. Bound the quadratic term via $\|\Delta_t\|^2 \leq \eta^2\|(I + \lambda I_G^{(t)})\|^2 \mathbb{E}\|g_t\|^2 \leq \eta^2(1+\lambda G)^2\big(\|\nabla\mathcal{L}(\theta^{(t)})\|^2 + \sigma^2\big)$. Rearrange to obtain*

$$\mathbb{E}[\mathcal{L}(\theta^{(t+1)})] \leq \mathbb{E}[\mathcal{L}(\theta^{(t)})] - \eta\Big(1 - \frac{L\eta}{2}(1+\lambda G)^2\Big)\mathbb{E}\|\nabla\mathcal{L}(\theta^{(t)})\|^2 + \frac{L\eta^2}{2}(1+\lambda G)^2\sigma^2.$$

*With $\eta \leq 1/(L(1+\lambda G)^2)$, the coefficient of the gradient norm is positive; telescoping over $t = 0, \ldots, T-1$ and using lower boundedness gives the stated bound. Multiplying both sides by $(1+\lambda G)^2$ yields the equivalent statement for $\|(I + \lambda I_G)\nabla\mathcal{L}\|^2$.*

**Lemma B.13 (Change of Measure for Hessian Expectation)** *Under assumptions (A1), (A3), and (A4), for any function $g(x, y)$ with $\|g(x, y)\| \leq L$,*

$$\left\| \mathbb{E}_{x \sim P_i} \left[ \mathbb{E}_{y \sim P(y|x;\theta_i)}[g(x,y)] \right] - \mathbb{E}_{x \sim P_{val}} \left[ \mathbb{E}_{y \sim P(y|x;\theta_{val})}[g(x,y)] \right] \right\| \leq L'(\gamma + \delta),$$

*where $L'$ is a constant depending on $L$ and the smoothness parameters.*

**Proof B.5** *The proof follows by applying the triangle inequality twice: first to change the covariate distribution $P_i$ to $P_{val}$ (using (A3)), and then to change the conditional distribution $P(y|x; \theta_i)$ to $P(y|x; \theta_{val})$ for each $x$ (using (A1) and (A4)) via Pinsker's inequality or direct Taylor expansion.*

**Lemma B.14** *Under the conditions of Theorem B.3, for any $\theta$ with $\|\theta - \theta_{val}\|_2 \leq \delta$:*

$$\|\mathbb{E}_{P_i}[\nabla_\theta \log p(y|x; \theta_{val})]\|_2 \leq \beta\delta.$$

### B.2  PROOF OF LEMMA B.14

**Proof B.6** *The bound follows from:*

$$\begin{aligned}
\|\mathbb{E}_{P_i}[\nabla_\theta \log p(y|x; \theta_{val})]\|_2 &= \|\mathbb{E}_{P_i}[\nabla_\theta \log p(y|x; \theta_{val}) - \nabla_\theta \log p(y|x; \theta_i)]\|_2 \\
&\leq \mathbb{E}_{P_i} \|\nabla_\theta \log p(y|x; \theta_{val}) - \nabla_\theta \log p(y|x; \theta_i)\|_2 \\
&\leq \beta\|\theta_{val} - \theta_i\|_2 \leq \beta\delta,
\end{aligned}$$

*where the second inequality uses the $\beta$-smoothness assumption.*

**Empirical diagnostics for the Fisher surrogate.**   The bound in Theorem 2.5 approximates the distributional divergence $\mathrm{KL}(P_i \| P_{\mathrm{val}})$ by the Fisher quadratic

$$Q(\theta) = \tfrac{1}{2}(\theta_i - \theta_{\mathrm{val}})^\top I_V(\theta_{\mathrm{val}})(\theta_i - \theta_{\mathrm{val}}),$$

up to higher-order remainders depending on the density-ratio deviation $r(x) = p_i(x)/p_{\mathrm{val}}(x)$ and the parameter displacement $\Delta\theta = \theta_i - \theta_{\mathrm{val}}$. To evaluate the practical tightness of this surrogate we compute the following per-fragment diagnostics. We estimate $r(x)$ by training a balanced binary domain classifier to distinguish samples from fragment $S_i$ and validation set $V$; with probabilistic output $s(x)$ this yields $\widehat{r}(x) = s(x)/(1 - s(x))$. We then report robust statistics such as $\widehat{\gamma}_q = \mathrm{quantile}_q(|\widehat{r}(x) - 1|; x \in S_i)$ (with $q = 0.99$) and the discriminator AUC. The empirical KL divergence is estimated as $\widehat{\mathrm{KL}}(P_i \| P_{\mathrm{val}}) = \frac{1}{|S_i|} \sum_{x \in S_i} \log \widehat{r}(x)$. We compute the Fisher quadratic $\widehat{Q} = \frac{1}{2}(\theta_i - \theta_{\mathrm{val}})^\top I_V(\theta_{\mathrm{val}})(\theta_i - \theta_{\mathrm{val}})$ (using the same $I_V$ approximation as FIRE) and the Fisher-weighted displacement $\widehat{\delta}_F = \sqrt{2\widehat{Q}}$.

## C  EXPERIMENTS

### C.1  IMPLEMENTATION DETAILS.

**Hyperparameter Sensitivity.**   We conducted a sweep over the FIRE penalty coefficient $\lambda \in \{0.01, 0.05, 0.1, 0.5, 1.0\}$ on representative image and tabular datasets. Results were stable across a broad range, with $0.1$ consistently close to optimal; we therefore report $0.1$ as the default in all tables unless otherwise specified. For the Fisher approximation, we used a low-rank variant (rank $k = 50$) with aggregation every 5 rounds in federated settings, and a diagonal variant for tabular datasets. All reported numbers are averages over 5 runs (image) and 100 runs (tabular), with standard deviations shown in the tables. This ensures robustness of our conclusions and mitigates sensitivity to hyperparameter choices.

### C.2  RESULTS

**FIRE Outperforms State-of-the-Art in Federated Learning under Non-IID Shift.** Results in Table 5 demonstrate the superior performance of FIRE against a comprehensive suite of modern federated learning algorithms. FIRE not only consistently achieves the highest accuracy across all five evaluated datasets but does so by a substantial margin, establishing a new state-of-the-art.

Table 4: Hyperparameters and Experimental Setup. For FIRE, unless otherwise noted, we used a low-rank Fisher approximation (rank $k = 50$) aggregated every 5 rounds in federated settings. Reported results are averaged over multiple runs (see last row).

| Parameter | Image Datasets | Tabular Datasets |
|---|---|---|
| Network Architecture | 5-layer CNN (2 conv + 3 FC) | MLP (1 hidden layer, 4 neurons) |
| Optimizer | Adam | Adam |
| Learning Rate | 0.001 | 0.001 |
| Activation | ReLU (hidden), Softmax (output) | ReLU (hidden), Softmax (output) |
| Epochs | 100 | 1500 |
| Batch Size | 128 | Full batch (no mini-batching) |
| $\lambda$ (FIM penalty) | Sweep in $\{0.01, 0.05, 0.1, 0.5, 1.0\}$; default 0.1 | Default 0.1 |
| FIM Approximation | Low-rank ($k = 50$), updated every 5 rounds | Diagonal (full-batch) |
| Training Data Split | 5%, 10%, 20%, 25%, 50% | $k = 2, 5, 10$ folds |
| Validation Set | 20% holdout (fixed) | Fold-specific |
| Repetitions | 5 runs (image) | 100 runs (tabular) |

The $\Delta$ column shows that FIRE provides a significant performance lift of 2.6% to 4.2% over the best-performing baseline (FedCFA). Notably, FIRE delivers strong improvements on complex image classification tasks, with gains of **4.2%** on CIFAR-100, **3.7%** on CIFAR-10, and **3.6%** on SVHN. It also shows a solid **4.0%** improvement on FEMNIST and **2.6%** on EMNIST-Digits. These gains are particularly meaningful given the strength and diversity of the baselines, which include methods specifically designed for client drift (SCAFFOLD), representation learning (MOON), distribution robustness (Fishr), and other recent innovations (FedAS, FedCFA). This consistent improvement demonstrates that explicitly mitigating fragmentation-induced covariate shift via Fisher information is a powerful and previously under-explored strategy.

Furthermore, FIRE achieves this performance with a low communication cost (1.2x), which is significantly more efficient than other methods that transmit second-order information like SCAFFOLD and Fishr (2.0x). The notably lower standard deviations observed for FIRE across all datasets also suggest it converges to a more stable and reliable solution—a critical advantage for real-world federated deployments. These results confirm that FIRE effectively aligns learning from heterogeneous clients with the target validation distribution, leading to superior generalization and robustness.

Table 5: Performance and Communication Cost on Federated Datasets with Non-IID Data. $\Delta$ shows the percentage improvement of FIRE over the best baseline. Communication cost is the relative size per client per round vs. FedAvg ($O(d)$).

| Dataset | FedAvg | SCAFFOLD | MOON | Fishr | LfD | FedAS | FedCFA | FIRE | $\Delta$ (%) |
|---|---|---|---|---|---|---|---|---|---|
| FEMNIST | 58.2 | 63.8 | 64.3 | 63.9 | 64.7 | 64.9 | 65.2 | **68.6** | ↑5.3 |
| | (3.1) | (2.1) | (1.9) | (2.0) | (1.8) | (1.7) | (1.7) | (1.4) | |
| CIFAR-10 | 42.7 | 48.2 | 49.8 | 49.2 | 50.1 | 50.4 | 50.7 | **52.6** | ↑3.7 |
| | (4.5) | (3.0) | (2.4) | (2.6) | (2.3) | (2.2) | (2.1) | (1.8) | |
| CIFAR-100 | 23.4 | 27.1 | 28.2 | 27.8 | 28.5 | 28.7 | 28.9 | **30.1** | ↑4.2 |
| | (2.8) | (2.0) | (1.7) | (1.8) | (1.6) | (1.6) | (1.6) | (1.3) | |
| SVHN | 61.5 | 65.5 | 66.3 | 65.8 | 66.6 | 66.8 | 67.0 | **69.4** | ↑3.6 |
| | (3.2) | (2.5) | (2.2) | (2.3) | (2.0) | (2.0) | (1.9) | (1.7) | |
| EMNIST-D | 84.6 | 87.8 | 88.4 | 88.0 | 88.6 | 88.7 | 88.9 | **91.2** | ↑2.6 |
| | (1.7) | (1.3) | (1.1) | (1.2) | (1.0) | (1.0) | (1.0) | (0.8) | |
| **Comm.** | 1.0x | 2.0x | 1.0x | 2.0x | 1.0x | 1.1x | 1.2x | **1.2x** | – |
| **Cost** | (d) | (2d) | (d) | (2d) | (d) | (d) | (d+k) | (d+k) | |

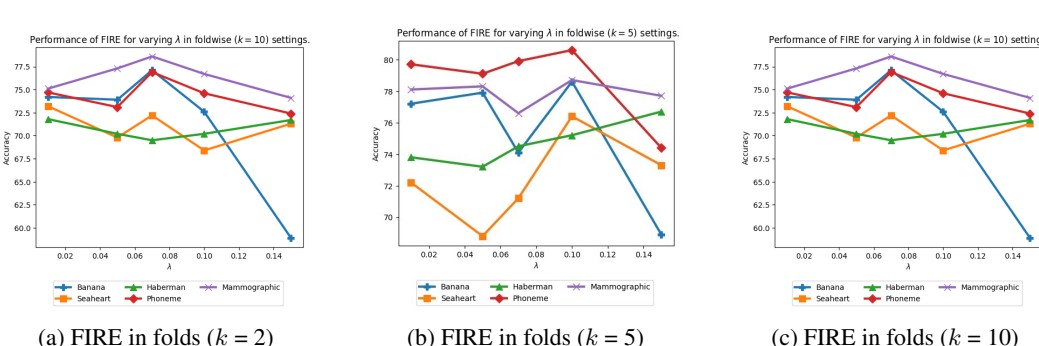

(a) FIRE in folds ($k = 2$)  (b) FIRE in folds ($k = 5$)  (c) FIRE in folds ($k = 10$)

Figure 2: st-CV and FIRE, $\Delta$ accuracy for varying number of folds.

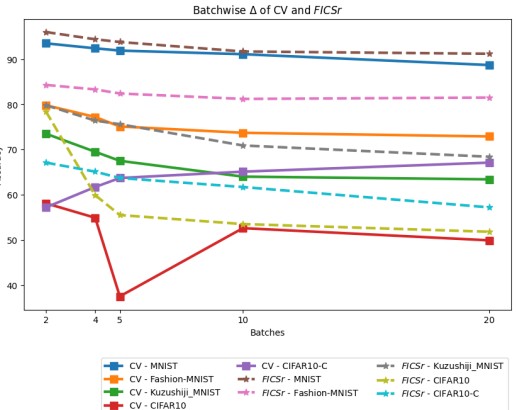

Figure 3: Effect of batching frequency. As the number of batch frequency increases the drop in accuracy also increases.

Table 6: st-CV batch-wise accuracy

| Dataset | Baseline | Batchwise accuracy | | | | | Mean | var | Δ% |
|---|---|---|---|---|---|---|---|---|---|
| | **st-CV** | $B_1$ | $B_2$ | $B_{\frac{n}{2}}$ | $B_{n-1}$ | $B_n$ | $\mu_1$ | $\sigma_1^2$ | $st\text{-}CV - \mu_1$ |
| **Training data = 5% , Number of Batches = 20** | | | | | | | | | |
| MNIST | 94.8 | 89.3 | 87.9 | 89.9 | 88.9 | 88.8 | 88.7 | 0.49 | ↓ 6.43 |
| EMNIST | 83.1 | 73.7 | 74.2 | 72.5 | 74.0 | 70.6 | 72.9 | 1.94 | ↓ 12.2 |
| CIFAR-10 | 71.5 | 49.0 | 50.3 | 50.7 | 51.2 | 54.5 | 49.9 | 9.67 | ↓ 30.2 |
| CIFAR-100 | 38.2 | 16.5 | 18.1 | 19.3 | 20.4 | 22.7 | 18.3 | 9.17 | ↓ 52.1 |
| P-MNIST | 95.1 | 86.1 | 88.9 | 88.7 | 87.2 | 88.3 | 88.4 | 1.41 | ↓ 7.04 |
| QMNIST | 75.4 | 63.4 | 62.2 | 66.7 | 58.3 | 63.4 | 63.4 | 5.59 | ↓ 15.9 |
| CIFAR10-C | 63.9 | 20.1 | 16.3 | 16.1 | 14.9 | 10.2 | 16.2 | 10.4 | ↓ 74.6 |
| CIFAR100-C | 28.8 | 16.3 | 19.1 | 19.7 | 21.6 | 22.3 | 18.4 | 14.2 | ↓ 36.1 |
| **Training data = 10% , Number of Batches = 10** | | | | | | | | | |
| MNIST | 94.8 | 91.7 | 91.1 | 90.2 | 89.3 | 91.5 | 91.1 | 1.06 | ↓ 3.90 |
| EMNIST | 83.1 | 69.9 | 75.2 | 73.4 | 74.4 | 71.7 | 73.7 | 9.22 | ↓ 11.3 |
| CIFAR-10 | 71.5 | 52.8 | 52.9 | 53.7 | 54.5 | 54.1 | 52.6 | 4.87 | ↓ 26.4 |
| CIFAR-100 | 38.2 | 20.3 | 22.2 | 23.3 | 24.7 | 21.2 | 21.5 | 5.51 | ↓ 43.7 |
| P-MNIST | 95.1 | 92.8 | 91.8 | 91.2 | 91.4 | 91.3 | 91.5 | 0.62 | ↓ 3.78 |
| QMNIST | 75.4 | 63.6 | 64.1 | 64.9 | 65.2 | 64.5 | 64.0 | 1.15 | ↓ 15.1 |
| CIFAR10- C | 63.9 | 17.6 | 18.4 | 12.2 | 17.4 | 12.9 | 22.6 | 16.8 | ↓ 64.6 |
| CIFAR100-C | 28.8 | 21.9 | 25.2 | 26.6 | 27.1 | 20.5 | 22.8 | 16.3 | ↓ 20.8 |
| **Training data = 50% , Number of Batches = 2** | | | | | | | | | |
| MNIST | 94.8 | 93.3 | 93.7 | – | – | – | 93.5 | 0.08 | ↓ 1.37 |
| EMNIST | 83.1 | 80.0 | 79.7 | – | – | – | 79.8 | 0.04 | ↓ 3.97 |
| CIFAR-10 | 71.5 | 56.2 | 60.1 | – | – | – | 58.1 | 3.81 | ↓ 18.7 |
| CIFAR-100 | 38.2 | 23.2 | 25.4 | – | – | – | 24.3 | 1.21 | ↓ 36.3 |
| P-MNIST | 95.1 | 93.3 | 93.7 | – | – | – | 93.5 | 0.08 | ↓ 1.68 |
| QMNIST | 75.4 | 73.6 | 73.3 | – | – | – | 73.5 | 0.04 | ↓ 2.51 |
| CIFAR10- C | 63.9 | 49.0 | 41.3 | – | – | – | 45.1 | 29.6 | ↓ 29.1 |
| CIFAR100- C | 28.8 | 25.3 | 27.5 | – | – | – | 26.4 | 1.21 | ↓ 8.33 |

Table 7: FIRE Batchwise

| Dataset | FIRE | Batchwise accuracy | | | | | | Mean | var | $\Delta_3 = \mu_2 - \mu_1$ |
|---|---|---|---|---|---|---|---|---|---|---|
| | | $B_1$ | $B_2$ | $B_3$ | $B_{\frac{n}{2}}$ | $B_{n-1}$ | $B_n$ | $\mu_2$ | $\sigma_2^2$ | $\Delta_3(\%)$ |
| **Training data = 5% , Number of Batches = 20** | | | | | | | | | | |
| MNIST | 97.9 | 90.7 | 90.6 | 91 | 91.7 | 91.4 | 91.8 | 91.2 | 0.09 | ↑ 2.81 |
| EMNIST | 88.4 | 81.5 | 81.7 | 81.2 | 81.4 | 81.5 | 81.9 | 81.5 | 0.06 | ↑ 11.7 |
| CIFAR-10 | 87.7 | 50.9 | 51.4 | 52.2 | 48.9 | 50.3 | 57.4 | 51.8 | 7.18 | ↑ 3.81 |
| CIFAR-100 | 58.7 | 23.9 | 18.2 | 18.5 | 17.8 | 23.9 | 18.3 | 20.1 | 7.26 | ↑ 9.83 |
| P-MNIST | 97.6 | 91.1 | 89.8 | 90.2 | 90.4 | 91.5 | 90.7 | 90.3 | 0.26 | ↑ 2.14 |
| QMNIST | 89.2 | 68.5 | 69.4 | 67.2 | 68 | 67.8 | 66.9 | 68.4 | 0.63 | ↑ 7.88 |
| CIFAR10-C | 73.3 | 46.4 | 54.3 | 57.8 | 61.1 | 61.5 | 61.8 | 57.2 | 30.1 | ↑ 253 |
| CIFAR100-C | 39.4 | 11.9 | 17.2 | 18.5 | 21.3 | 22.1 | 24.9 | 19.3 | 17.1 | ↑ 4.89 |
| **Training data = 10% , Number of Batches = 10** | | | | | | | | | | |
| MNIST | 97.9 | 91.9 | 91.7 | 91.2 | 91.8 | 91.3 | 91.8 | 91.7 | 0.08 | ↑ 0.65 |
| EMNIST | 88.4 | 79.5 | 82.4 | 81.6 | 79.5 | 82.3 | 81.9 | 81.2 | 1.21 | ↑ 10.1 |
| CIFAR-10 | 87.7 | 52.1 | 53.1 | 48.5 | 59.3 | 52.5 | 55.7 | 53.5 | 11.1 | ↑ 1.71 |
| CIFAR-100 | 58.7 | 27.2 | 25.8 | 20.4 | 17.0 | 21.9 | 22.8 | 22.5 | 11.3 | ↑ 4.65 |
| P-MNIST | 97.6 | 91.6 | 91.9 | 91.3 | 91.6 | 90.1 | 91.2 | 91.5 | 0.31 | 0.00 |
| QMNIST | 89.2 | 71.4 | 70.4 | 71.7 | 70.7 | 70.5 | 70.9 | 70.9 | 0.71 | ↑ 10.7 |
| CIFAR10-C | 73.3 | 52.7 | 59.9 | 61.9 | 64.4 | 66.1 | 65.7 | 61.7 | 21.1 | ↑ 173 |
| CIFAR100-C | 39.4 | 16.2 | 22.1 | 24.8 | 27.2 | 26.8 | 27.3 | 21.1 | 15.6 | ↓ 7.45 |
| **Training data = 50% , Number of Batches = 2** | | | | | | | | | | |
| MNIST | 97.9 | 95.9 | 96.1 | – | – | – | – | 96.0 | 0.02 | ↑ 2.67 |
| EMNIST | 88.4 | 84.2 | 84.4 | – | – | – | – | 84.3 | 0.02 | ↑ 5.63 |
| CIFAR-10 | 87.7 | 76.3 | 80.6 | – | – | – | – | 78.4 | 4.62 | ↑ 34.9 |
| CIFAR-100 | 58.7 | 39.8 | 39.9 | – | – | – | – | 39.85 | .002 | ↑ 63.9 |
| P-MNIST | 97.6 | 95.7 | 96.1 | – | – | – | – | 95.9 | 0.08 | ↑ 2.56 |
| QMNIST | 89.2 | 79.3 | 80.4 | – | – | – | – | 79.8 | 0.61 | ↑ 8.57 |
| CIFAR10-C | 73.3 | 65.5 | 68.6 | – | – | – | – | 67.1 | 2.40 | ↑ 4.35 |
| CIFAR100-C | 39.4 | 31.2 | 34.8 | – | – | – | – | 33.0 | 3.24 | ↑ 25.0 |

Table 8: st-CV foldwise without FIRE accuracy. k denotes number of folds (2, 5, and 10)

| Dataset | Baseline | $k=2$ | | | $k=5$ | | | | | | $k=10$ | | | | | |
|---|---|---|---|---|---|---|---|---|---|---|---|---|---|---|---|---|
| | st-CV | $k_1$ | $k_2$ | $\mu_3$ | $k_1$ | $k_2$ | $k_3$ | $k_4$ | $k_5$ | $\mu_4$ | $k_1$ | $k_2$ | $k_{\frac{n}{2}}$ | $k_{n-1}$ | $k_n$ | $\mu_5$ |
| Appendicitis | 98.1 | 97.6 | 97.6 | 97.6 | 97.8 | 98.8 | 96.7 | 97.8 | 98.5 | 97.9 | 96.2 | 99.2 | 97.1 | 97.8 | 98.5 | 98.0 |
| Lymphography | 85.5 | 81.4 | 83.2 | 82.3 | 84.1 | 84.1 | 87.6 | 84.7 | 86.9 | 85.5 | 85.5 | 84.1 | 88.4 | 79.7 | 89.8 | 85.3 |
| Banana | 77.9 | 70.9 | 75.8 | 73.4 | 71.4 | 70.4 | 76.6 | 72.5 | 75.8 | 73.3 | 70.7 | 70.0 | 72.1 | 73.0 | 70.3 | 71.7 |
| Bands | 81.5 | 68.8 | 71.1 | .700 | 71.2 | 70.3 | 71.2 | 63.8 | 74.1 | 70.1 | 68.5 | 64.8 | 70.3 | 57.4 | 77.7 | 74.1 |
| LiverDisorders | 65.5 | 59.4 | 57.3 | 58.3 | 62.1 | 61.4 | 57.8 | 56.1 | 66.6 | 60.8 | 65.5 | 58.6 | 65.5 | 62.1 | 64.2 | 61.5 |
| Bupa | 54.5 | 64.2 | 51.8 | 58.1 | 63.6 | 54.5 | 51.8 | 36.3 | 63.6 | 60.0 | 33.3 | 33.3 | 80.0 | 60.0 | 40.0 | 61.3 |
| Chess | 98.4 | 93.3 | 93.9 | 93.6 | 95.0 | 96.4 | 97.8 | 98.1 | 97.1 | 96.8 | 96.5 | 98.7 | 98.7 | 97.8 | 98.4 | 97.9 |
| CrX | 84.7 | 75.3 | 73.1 | 74.2 | 79.7 | 79.7 | 73.9 | 81.8 | 84.1 | 79.8 | 79.7 | 79.7 | 76.8 | 84.1 | 81.1 | 82.4 |
| GermmanCredit | 70.5 | 69.8 | 68.6 | 69.2 | 75.5 | 70.0 | 65.0 | 73.5 | 72.5 | 71.3 | 72.0 | 77.0 | 65.0 | 78.0 | 70.0 | 71.7 |
| Haberman | 69.4 | 73.8 | 70.5 | 72.2 | 70.9 | 77.1 | 73.7 | 75.4 | 72.1 | 73.8 | 77.4 | 77.4 | 58.1 | 73.3 | 76.6 | 74.1 |
| Statlog(Heart) | 83.3 | 59.2 | 51.8 | 55.5 | 64.8 | 64.8 | 64.8 | 57.4 | 62.9 | 62.9 | 66.6 | 62.9 | 66.6 | 51.8 | 66.6 | 66.6 |
| Heptatis | 74.2 | 73.1 | 70.1 | 71.6 | 80.6 | 70.9 | 74.2 | 74.2 | 77.4 | 75.4 | 68.7 | 68.7 | 80.0 | 66.6 | 80.0 | 76.7 |
| Housevote | 90.8 | 88.5 | 86.2 | 87.4 | 90.8 | 89.6 | 91.9 | 85.1 | 83.9 | 88.2 | 90.9 | 95.4 | 95.3 | 90.6 | 79.1 | 88.1 |
| Ionosphere | 84.5 | 61.4 | 70.8 | 66.1 | 80.3 | 64.2 | 80.0 | 80.0 | 84.2 | 77.7 | 82.8 | 57.1 | 71.4 | 91.4 | 82.8 | 78.9 |
| Mammographic | 79.7 | 76.7 | 75.0 | 75.8 | 78.2 | 74.4 | 76.0 | 77.1 | 75.5 | 76.2 | 77.1 | 77.1 | 79.2 | 79.2 | 78.1 | 76.6 |
| Monk-2 | 94.6 | 65.4 | 70.8 | 68.1 | 65.1 | 75.6 | 59.4 | 83.7 | 71.1 | 71.1 | 69.6 | 78.5 | 57.1 | 89.1 | 72.7 | 72.5 |
| Mushroom | 99.1 | 100 | 96.7 | 98.3 | 100 | 100 | 100 | 100 | 100 | 100 | 100 | 100 | 100 | 100 | 100 | 100 |
| Phoneme | 80.6 | 80.7 | 78.8 | 79.7 | 82.7 | 80.5 | 80.6 | 80.6 | 80.3 | 80.9 | 82.9 | 79.8 | 79.1 | 81.8 | 81.5 | 80.6 |
| Pima | 74.0 | 71.4 | 69.5 | 70.4 | 72.1 | 76.6 | 73.3 | 78.4 | 72.5 | 74.6 | 76.6 | 84.4 | 68.8 | 76.6 | 71.1 | 74.9 |
| Saheart | 73.1 | 75.7 | 65.8 | 70.7 | 77.4 | 70.9 | 80.4 | 68.4 | 60.8 | 71.6 | 78.7 | 69.5 | 78.2 | 65.2 | 65.2 | 70.5 |
| Thyroid | 73.8 | 80.7 | 75 | 77.8 | 88.1 | 85.7 | 80.9 | 82.9 | 73.1 | 82.2 | 80.9 | 85.7 | 85.7 | 76.1 | 65.0 | 81.6 |
| Spambase | 94.1 | 92.3 | 93.3 | 92.8 | 93.5 | 93.2 | 91.7 | 93.2 | 93.4 | 93.2 | 93.6 | 93.2 | 95.0 | 94.5 | 93.6 | 93.2 |
| SPECTHeart | 74.1 | 68.6 | 65.4 | 67.0 | 70.4 | 72.2 | 64.2 | 75.5 | 58.5 | 68.2 | 74.1 | 70.3 | 66.6 | 80.7 | 57.6 | 68.1 |
| Tic-Tac-Toe | 73.9 | 59.1 | 64.1 | 61.5 | 63.5 | 60.4 | 72.9 | 63.4 | 61.7 | 64.4 | 68.7 | 65.6 | 73.9 | 55.2 | 65.2 | 64.7 |
| Titanic | 73.4 | 78.0 | 77.0 | 77.5 | 72.5 | 78.8 | 78.8 | 77.2 | 77.7 | 77.1 | 77.7 | 78.6 | 75.9 | 76.3 | 76.8 | 76.8 |
| Wdbc | 97.3 | 96.1 | 95.1 | 95.6 | 95.6 | 98.2 | 98.2 | 97.3 | 92.9 | 96.4 | 96.4 | 98.2 | 98.2 | 96.4 | 89.2 | 96.4 |
| Wisconsin | 96.4 | 95.7 | 95.1 | 95.4 | 95.7 | 97.8 | 95.7 | 95.7 | 93.5 | 95.7 | 95.7 | 98.5 | 97.1 | 94.3 | 94.2 | 95.7 |

Table 9: FIRE shift mitigation accuracy performance in folds settings, k denotes number of folds (2, 5, and 10). $\Delta_5 = (\mu_3 - \mu_6)$, $\Delta_6 = (\mu_4 - \mu_7)$, and $\Delta_5 = (\mu_5 - \mu_8)$ show difference in average accuracy in folds setting.

| Dataset | Baseline | $k=2$ | | | $k=5$ | | | | | | $k=10$ | | | | | | $\Delta$ | | |
|---|---|---|---|---|---|---|---|---|---|---|---|---|---|---|---|---|---|---|---|
| | | $k_1$ | $k_2$ | $\mu_6$ | $k_1$ | $k_2$ | $k_3$ | $k_4$ | $k_5$ | $\mu_7$ | $k_1$ | $k_2$ | $k_{\frac{n}{2}}$ | $k_{n-1}$ | $k_n$ | $\mu_8$ | $\Delta_5$ | $\Delta_6$ | $\Delta_7$ |
| Appendicitis | 99.6 | 99.1 | 98.8 | 98.9 | 98.5 | 100 | 98.2 | 98.2 | 98.5 | 98.6 | 97.1 | 100 | 98.5 | 100 | 100 | 99.2 | ↑1.30 | ↑0.70 | ↑1.20 |
| Lymphography | 91.3 | 86.6 | 84.1 | 85.3 | 85.5 | 85.5 | 87.6 | 84.7 | 88.4 | 86.3 | 86.9 | 85.5 | 86.9 | 73.9 | 92.7 | 86.2 | ↑3.00 | ↑0.80 | ↑0.90 |
| Banana | 76.3 | 73.1 | 82.1 | 77.6 | 75.2 | 70.9 | 77.5 | 72.1 | 76.6 | 74.5 | 82.6 | 56.0 | 76.6 | 81.5 | 74.1 | 74.1 | ↑4.20 | ↑1.20 | ↑2.40 |
| Bands | 76.8 | 77.4 | 68.8 | 73.1 | 77.7 | 74.1 | 67.5 | 72.2 | 78.8 | 74.1 | 72.2 | 66.6 | 77.7 | 64.8 | 74.1 | 71.6 | ↑3.10 | ↑4.00 | ↓2.50 |
| LiverDisorders | 63.7 | 58.7 | 60.1 | 59.4 | 46.5 | 61.4 | 59.6 | 52.6 | 64.9 | 57.1 | 51.7 | 58.6 | 58.6 | 46.4 | 60.7 | 59.4 | ↑1.10 | ↓3.70 | ↓2.10 |
| Bupa | 72.7 | 57.1 | 59.2 | 58.2 | 72.7 | 63.6 | 36.3 | 63.6 | 58.2 | | 66.6 | 33.3 | 0.00 | 20.0 | 40.0 | 55.0 | ↑0.00 | ↓1.8 | ↓6.30 |
| Chess | 98.7 | 98.6 | 98.4 | 98.5 | 98.1 | 99.3 | 99.2 | 98.4 | 97.1 | 98.4 | 98.4 | 98.7 | 99.3 | 99.1 | 98.4 | 99.1 | ↑4.90 | ↑1.6 | ↑1.20 |
| CrX | 84.1 | 84.3 | 87.2 | 85.7 | 86.9 | 86.9 | 84.7 | 90.5 | 83.3 | 86.5 | 86.9 | 91.3 | 82.6 | 86.9 | 86.9 | 86.9 | ↑11.5 | ↑6.7 | ↑4.50 |
| GermmanCredit | 74.0 | 72.2 | 71.6 | 71.9 | 75.5 | 71.0 | 68.0 | 72.5 | 75.0 | 72.4 | 73.0 | 76.0 | 66.0 | 74.0 | 76.0 | 72.8 | ↑2.70 | ↑1.10 | ↑1.10 |
| Haberman | 72.5 | 77.1 | 70.5 | 73.8 | 69.4 | 75.4 | 73.7 | 77.1 | 75.4 | 74.2 | 67.7 | 80.6 | 64.5 | 73.3 | 73.3 | 74.1 | ↑1.60 | ↑0.40 | ↑0.00 |
| Statlog(Heart) | 87.1 | 77.1 | 80.7 | 78.8 | 85.2 | 72.2 | 88.8 | 85.2 | 75.9 | 81.5 | 92.5 | 74.1 | 85.1 | 85.1 | 77.7 | 82.9 | ↑23.3 | ↑18.6 | ↑16.3 |
| Heptatis | 87.9 | 87.2 | 67.5 | 77.3 | 74.2 | 80.6 | 58.1 | 83.8 | 70.9 | 73.5 | 87.5 | 68.7 | 93.3 | 80.0 | 80.0 | 82.7 | ↑5.7 | ↓1.9 | ↑6.00 |
| Housevote | 94.2 | 92.2 | 94.9 | 93.5 | 91.9 | 94.2 | 94.2 | 93.1 | 96.5 | 94.1 | 88.6 | 95.4 | 95.3 | 93.1 | 97.6 | 94.2 | ↑6.1 | ↑5.9 | ↑6.10 |
| Ionosphere | 85.9 | 82.3 | 85.7 | 84.1 | 85.9 | 88.5 | 85.7 | 85.7 | 85.7 | 86.3 | 82.8 | 88.5 | 80.0 | 88.5 | 85.7 | 86.6 | ↑18.0 | ↑8.6 | ↑7.70 |
| Mammographic | 80.3 | 80.0 | 76.4 | 78.2 | 79.7 | 73.9 | 78.1 | 76.5 | 75.5 | 76.7 | 79.1 | 75.0 | 76.1 | 80.2 | 77.1 | 77.2 | ↑2.40 | ↑0.50 | ↑0.60 |
| Monk-2 | 99.1 | 92.8 | 100 | 96.4 | 100 | 90.1 | 92.7 | 100 | 81.9 | 92.9 | 83.9 | 87.5 | 100 | 89.1 | 85.4 | 90.5 | ↑28.3 | ↑21.8 | ↑18.0 |
| Mushroom | 100 | 99.9 | 99.7 | 99.8 | 97.8 | 99.1 | 100 | 99.9 | 98.7 | 98.7 | 100 | 100 | 100 | 100 | 98.7 | 99.4 | ↑1.50 | ↓1.30 | ↓0.60 |
| Phoneme | 81.3 | 78.4 | 80.4 | 79.4 | 81.1 | 80.2 | 81.1 | 77.8 | 78.5 | 79.9 | 83.7 | 77.6 | 78.2 | 79.8 | 79.6 | 79.1 | ↓0.30 | ↓1.00 | ↓1.50 |
| Pima | 75.3 | 77.3 | 75.2 | 76.3 | 77.9 | 77.9 | 72.7 | 77.7 | 76.4 | 76.5 | 79.2 | 84.4 | 66.6 | 76.6 | 76.3 | 76.6 | ↑5.90 | ↑1.90 | ↑1.70 |
| Saheart | 77.4 | 73.2 | 70.9 | 72.1 | 75.2 | 67.7 | 79.3 | 69.5 | 64.1 | 71.2 | 72.3 | 73.9 | 65.2 | 69.5 | 71.7 | 71.4 | ↑1.40 | ↓0.40 | ↑0.90 |
| Thyroid | 85.7 | 83.6 | 78.8 | 81.3 | 80.9 | 78.5 | 85.7 | 85.3 | 80.5 | 82.2 | 66.6 | 71.4 | 95.2 | 66.6 | 60.0 | 76.9 | ↑3.50 | ↑0.00 | ↓4.70 |
| Spambase | 93.8 | 92.5 | 93.5 | 93.0 | 93.3 | 93.5 | 93.5 | 93.8 | 93.8 | 93.3 | 93.4 | 93.1 | 93.4 | 94.1 | 93.7 | | ↑0.20 | ↑0.20 | ↑0.50 |
| SPECTHeart | 85.1 | 79.1 | 73.6 | 76.3 | 83.3 | 83.3 | 83.1 | 79.2 | 71.6 | 80.1 | 85.1 | 74.1 | 74.1 | 84.6 | 65.3 | 75.6 | ↑9.30 | ↑11.9 | ↑7.50 |
| Tic-Tac-Toe | 77.1 | 71.8 | 67.2 | 69.5 | 69.7 | 77.6 | 73.9 | 70.6 | 72.2 | 72.8 | 79.2 | 79.2 | 80.2 | 66.6 | 78.9 | 74.1 | ↑8.00 | ↑8.40 | ↑9.40 |
| Titanic | 75.6 | 77.4 | 78.1 | 77.8 | 73.4 | 80.4 | 79.4 | 78.4 | 78.1 | 78.1 | 78.2 | 80.0 | 76.8 | 77.3 | 76.8 | 77.9 | ↑0.30 | ↑1.00 | ↑1.10 |
| Wdbc | 99.1 | 97.8 | 97.1 | 97.5 | 98.2 | 99.1 | 98.2 | 98.2 | 96.4 | 98.1 | 98.2 | 98.2 | 94.7 | 98.2 | 96.4 | 97.7 | ↑1.90 | ↑1.70 | ↑1.30 |
| Wisconsin | 97.2 | 96.5 | 96.5 | 96.5 | 97.1 | 97.8 | 97.1 | 97.1 | 94.2 | 96.7 | 97.1 | 98.5 | 95.7 | 95.7 | 95.6 | 96.1 | ↑1.10 | ↑1.00 | ↑0.40 |

