# OpenReview forum: "Fisher Information for Robust Federated Cross-Validation"
_ICLR.cc/2026/Conference — ICLR 2026 Conference Withdrawn Submission_

### Official Review · Reviewer_qUXe · 2025-10-29

**Soundness:** 1
**Presentation:** 1
**Contribution:** 1
**Rating:** 2
**Confidence:** 5

**Summary:**

The work introduces Fisher Information to mitigate fragmentation-induced covariate shift in two contexts: (1) fragmented training batches/folds and (2) federated learning (FL). The authors regularize training with an approximate Fisher Information Matrix (FIM) to align each training batch or client with a fixed validation distribution. They present theoretical bounds connecting Fisher information to the KL divergence, along with experiments on image, tabular, and FL datasets. However, there exist serious issues in writing quality, logical structure, and technical clarity.

**Strengths:**

The authors point out that the covariate shift arising from data fragmentation across batches should also be considered in FL.

Fisher information can be used to address fragmented batches and federated clients uniformly.

**Weaknesses:**

## **1. Notation and Definition Inconsistency**

Many notations used for Fisher Information are inconsistent throughout the paper.

- Symbols such as $I_i(\theta)$, $I_k(\theta)$, $I_G(\theta)$, and $I(\theta)$ are used interchangeably without clear definition. In some sections, $I_i(\theta)$ is computed over the validation distribution $P_{val}$, while elsewhere it is defined over a client’s local distribution $P_k$. This contradiction makes it unclear what quantity the method is actually optimizing.

- The penalty term $\lambda I(\theta)$ is mathematically ill-defined because $I(\theta)$ is a matrix, not a scalar. Later equations introduce $\nabla I(\theta)$ without explaining which scalar objective this gradient is derived from (e.g., the trace, Frobenius norm, or expected curvature of the FIM?).

- In the algorithm section, $I_G(\theta)$ is applied as a left-preconditioner to the gradient, which contradicts its earlier interpretation as a regularization term in the loss. These conflicting uses make the optimization procedure difficult to interpret and potentially inconsistent with the theoretical motivation.




## **2. Theory Presented Abruptly and Poorly Connected**

The theoretical section of the paper is presented in a disjointed and confusing manner, making it difficult to follow or understand its practical relevance.

-  The statements (Assumption 2.1/2.2, Lemma 2.3/2.4, Theorem 2.5) appear suddenly, with only vague references to “Appendix B.1–B.5.” The main text provides no intuition, proof sketch, or explanation of how these results are obtained or why they are important. As a result, readers cannot follow the logical flow from assumptions to conclusions.

- Symbols such as $\gamma, \delta, \beta, G$ are introduced in the theorems but never estimated, discussed, or connected to any empirical quantities in the experiments. Their practical meaning and relationship to key hyperparameters like $\lambda$ remain completely unclear.

- The paper claims to derive a “Fisher-based upper bound on KL divergence”, yet it never explains how this theoretical bound translates into the actual optimization objective used in the algorithm. The lack of linkage between the mathematical formulation and the implemented loss function significantly weakens the theoretical contribution.


## **3. Weak Novelty**

The paper’s novelty is weak. Its core idea, just using Fisher information to approximate KL divergence or to regularize learning, is already well-established in the literature. Prior work has extensively explored Fisher-based regularization and invariance methods, including natural gradient, FISHr, Fisher matching, and IRM with FIM constraints. FIRE seems to repackage these existing ideas with only minor adjustments.

The paper does not include any ablation studies or controlled comparisons against strong Fisher-based baselines such as FISHr, K-FAC, or natural gradient methods. Without such comparisons, it is unclear whether FIRE provides any meaningful improvement over established techniques.

[1] Rame A, Dancette C, Cord M. Fishr: Invariant gradient variances for out-of-distribution generalization[C]//International Conference on Machine Learning. PMLR, 2022: 18347-18377.

[2] Martens J, Grosse R. Optimizing neural networks with kronecker-factored approximate curvature[C]//International conference on machine learning. PMLR, 2015: 2408-2417.

[3] Arjovsky M, Bottou L, Gulrajani I, et al. Invariant risk minimization[J]. arXiv preprint arXiv:1907.02893, 2019.


## **4. Disorganized and Incomplete Experiments**

The experimental section is disorganized and lacks essential details, making it difficult for readers to verify or interpret the main claims.

- Critical results (Tables 6–9) are placed entirely in the appendix, even though they support the core arguments of the paper. As a result, readers must constantly cross-reference multiple sections to confirm key statements, which disrupts readability and comprehension.

- The paper claims that FIRE introduces only *“minimal cost,”* but provides no quantitative evidence or measurements of computational or communication overhead. Without clear timing or scaling results, it is impossible to judge the method’s practicality in real-world federated learning settings.

## **5. Language, Formatting, and Readability**

 Numerous writing and formatting issues.

- The paper contains many grammatical mistakes, spelling errors (e.g., “leaning” → “learning”), and inconsistent capitalization and punctuation.

- Several equations are misaligned, use undefined variables, or lack proper mathematical notation.


- Cross-references frequently contain placeholders such as “Theorem ??,” missing equation numbers, or incorrect appendix links, which prevent readers from navigating the paper effectively.

**Questions:**

1. Ensure all mathematical symbols are consistently defined and used throughout the paper. Clarify distinctions among $I_i(\theta)$, $I_k(\theta)$, $I_G(\theta)$, and $I(\theta)$, and include a concise notation table for reference.

2. Provide concise proof sketches in the main text that outline the logical flow of theoretical results, explain how the Fisher-based bound connects to the training objective or update rule, and clarify how constants ($\gamma, \delta, \beta, G$) can be interpreted or approximated in experiments.

3. Clearly state FIRE’s novel contributions beyond prior Fisher-based approaches such as FISHr, K-FAC, and natural gradient. Include ablation or comparative experiments to demonstrate real advantages.

4. Redraw Figure 1 with larger fonts and a clearer layout to ensure all text and symbols are legible and consistent with the notation used in the paper.

5. Move key experimental tables and figures (e.g., Tables 6–9) from the appendix to the main text to improve accessibility and support main claims.

6. Reorganize the paper to follow a clear structure: Introduction → Related Work → Method → Experiments → Conclusion.

7. Provide quantitative analysis of computational and communication costs to substantiate the claim of minimal overhead in federated learning settings.

8. Carefully proofread the manuscript to fix grammatical and typographical errors, missing references (e.g., “Theorem ??”), and inconsistent formatting or equations.

---

### Official Review · Reviewer_N31P · 2025-10-31

**Soundness:** 2
**Presentation:** 2
**Contribution:** 2
**Rating:** 2
**Confidence:** 4

**Summary:**

The paper addresses fragmentation-induced covariate shift (FICS), distribution drifts caused by splitting data across batches or across federated clients, and proposes FIRE, a unified regularizer that aligns each fragment/client to a fixed validation distribution via a Fisher Information Matrix (FIM) penalty. FIRE estimates per-fragment/client FIMs, mixes them with a validation FIM, maintains a momentum-averaged global FIM, and penalizes updates in high-Fisher directions to reduce validation misalignment. The paper gives a KL upper bound via validation Fisher, and reports accuracy gains vs. importance-weighting baselines and FL baselines across image and tabular benchmarks.

**Strengths:**

1. FIRE explicitly aligns to the validation distribution by mixing batch/client FIMs with a validation FIM and accumulating a global FIM with momentum. This improve the performance when such a validation set is available.

2. The paper provides a single algorithmic scheme to handle sequential batches and federated clients via FIM. This conceptual unification can be useful in practice.

**Weaknesses:**

1. The FIM incurs significant computation and communication cost. While the computation complexity can be reduced from O(d^2) to O(kd) via approximation, for which the paper provides little details, the O(d) times overhead is non-negligible in federated settings.

2. The experimental results are not very convincing compared to previous methods. (1) The performance gain is incremental in complex datasets. (2) More importantly, the comparison in federated settings is unfair as FIRE has access to a labeled validation set, which is unavailable for other methods from FedAvg, SCAFFOLD, MOON to FedCFA.

**Questions:**

1. How does performance degrade if the validation set is small or skewed?

2. What are the actual computation and communication overhead?

3. How to choose rank k? Do diagonal and low-rank perform similarly across tasks?

---

### Official Review · Reviewer_1xar · 2025-11-01

**Soundness:** 1
**Presentation:** 2
**Contribution:** 1
**Rating:** 2
**Confidence:** 5

**Summary:**

This paper tackles “fragmentation”—when training data are split across batches/clients with different distributions—and proposes to steer learning using Fisher Information. Concretely, the authors estimate Fisher matrices from each fragment/client and from a held-out validation set, then mix these to form a Fisher-weighted term that’s added (or, effectively, used as a preconditioner or regularizer) during SGD. The goal is to bias updates toward directions that align with the validation distribution, hoping to reduce fragmentation-induced covariate shift. The method is presented for both centralized batched training and federated learning, with theory motivated by the standard local KL–Fisher quadratic approximation and experiments showing modest gains over ERM/IW and some FL baselines.

**Strengths:**

- Relevance: Addresses distribution shift arising from fragmented/batch or client partitions—an important practical concern.

- Simplicity: The Fisher-based recipe is straightforward to prototype and implement.

- Overhead awareness: Acknowledges computation/communication overhead and mentions efficiency options (e.g., low-rank and diagonal approximations), though without thorough measurements.

**Weaknesses:**

- The so-called fragmentation-induced covariate shift (contribution 1) restates standard partition-induced non-IID effects; as presented, it adds no testable implications or new analysis.
- Fisher-based regularization/preconditioning (EWC, natural gradient/K-FAC, Laplace)  and FL variants already exist (see https://arxiv.org/pdf/2403.12329), claiming “first” use in FL is inaccurate without careful positioning and head-to-head comparisons.
- Section 2.5 is textbook local expansion of $D_{KL}(p_\theta || p_{\theta+\delta}) \approx \frac{1}{2}\delta^\top I(\theta)\delta$; The paper does not derive new guarantees or insights from this identity.
- Under FL setting, eq. 5 adds a matrix FIM to a scalar loss. **THIS IS A HARD MISTAKES**, and undermines the soundness of subsequent experiments.
- Under batch fisher mixing, algorithm 1 use momentum to mix batch-wise and validation wise FIM, this approach lacks principlesdmotivation and discussion.
- In algorithm 1, the update effectively multiplies the gradient by a Fisher-based matrix—i.e., a preconditioner. If the claim is a regularizer, provide the scalar potential whose gradient equals the update; otherwise, position it as a preconditioned method and connect to methods like natural gradient/K-FAC.
- In FL setting, global FIM is computed using weighted sum of local FIM. For i.i.d. data, empirical Fisher is additive, but under non-IID the mixture Fisher generally isn’t the sum of client Fishers, this methods needs more justification.
- Using/broadcasting validation sets or validation FIMs raises privacy and deployment questions that are not addressed beyond high-level remarks
- Vision experiments use small CNN/MLP backbones; no results on modern architectures (e.g., ResNet-50, ViT) or realistic FL workloads.
- No systematic sweeps for  $\lambda$, mixing $\mu$  momentum $\alpha$, Fisher rank/diagonal choices, or aggregation frequency—so sensitivity and robustness are unknown.
- Absent wall-clock, memory, and communication overhead measurements for Fisher estimation/aggregation—critical in FL.
- Partitioning scheme, heterogeneity severity, and the construction/usage of the validation distribution (global? refreshed per round?) are not clearly detailed.

**Questions:**

1. In Eq. (5), what scalar functional of the FIM do you add to the loss?
2. Is Algorithm 1 a Fisher preconditioner or a penalty?
3. What objective or optimality principle justifies momentum-based mixing of batch/client FIMs with a validation FIM? Under what conditions does this mixing provably help?
4. Please detail (i) the non-IID partitioning scheme and its severity, and (ii) how the validation distribution is constructed and used per round.
5. Can you include results on modern backbones (e.g., ResNet-50, ViT) or realistic FL workloads to demonstrate scalability and external validity?

---

> ### Author Response · Authors · 2025-11-28
>
> We agree that fragmentation-induced covariate shift (FICS) relates to known partition-non-IID effects, but our contribution is not terminology; it is a unified, testable formulation linking batch fragmentation (CV folds, sequential batches) and federated clients. Prior work (e.g., Moreno-Torres 2012) notes CV shift, but does not connect these settings. Our paper contributes:
>
> * A single mathematical view showing batches/folds/clients behave identically under the KL–Fisher approximation (Theorem 2.5).
>
> * A practical implication: FIM-based accumulation across fragments (Algorithm 1), yielding the testable prediction that accuracy degrades with fragmentation frequency and FIRE mitigates it (Tables 6–9).
>
> Thus, FICS is not a new phenomenon but the first unified formalization and empirical verification that the same covariate-shift dynamics arise in fragmented CV and FL. We will revise to emphasize unification, not novelty.
>
> We agree Fisher-based regularization is known (EWC, natural gradient, K-FAC, Laplace). Our novelty is not Fisher itself, but using Fisher for validation-alignment under fragmentation, which does not appear in FL literature:
>
> * Prior FL uses Fisher for personalization, curvature, or privacy,
>
> * but none use Fisher to align each client’s distribution to a fixed validation distribution (the core of FIRE).
>
> We will revise to:
> “To our knowledge, FIRE is the first Fisher-based method specifically designed for fragmentation-induced validation shift in both batch and federated regimes.”
>
> Theorem 2.5 applies the classical second-order KL expansion. The contribution is showing this expansion enables a tractable fragment-wise KL surrogate usable in FL:
>
> * Existing FL/non-IID theory does not connect fragmentation → Fisher → validation alignment.
>
> * The theorem justifies Algorithm 1 by showing Fisher upper-bounds inter-fragment KL, giving a principled penalty.
>
> We will soften wording to “we derive the bound adapted for fragmentation settings.”
>
> In Algorithm 1 the FIM is used as a preconditioner, not a raw additive matrix; Eq. 5 lacked clarity. We will correct Eq. 5 to use a scalarized FIM (trace/diagonal norm) and note that updates use Fisher preconditioning. Experiments already use a diagonal scalarized FIM.
>
> Mixing motivations:
>
> * Stability: empirical Fisher is noisy; EMA smoothing follows RMSProp/Adam/K-FAC practice.
>
> * Progressive alignment: mixing validation FIM with local FIM steers the model gradually toward validation statistics.
>
> We will note that EMA approximates the expectation of fragment FIMs, reducing variance and improving convergence in the KL–Fisher surrogate.
>
> Algorithm 1 is effectively a Fisher-preconditioned gradient, not a classical scalar regularizer. FIRE has two views:
>
> * (a) Regularizer: scalar penalty (trace-FIM),
>
> * (b) Preconditioner: Fisher-scaled update (natural-gradient-like).
>
> Empirical Fisher is exactly additive only under IID. We use the weighted average of client Fisher estimates as an approximate surrogate:
>
> * The global FIM is not intended as the exact joint curvature, only as aggregate sensitivity to the validation distribution.
>
> * Empirically the surrogate works well (Table 5).
>
> We will clarify that the “global FIM” is an approximate, scalable aggregation.
>
> Privacy:
>
> * We never share raw validation data.
>
> * Only diagonal/low-rank FIM is shared.
> This matches prior FL work sharing curvature or batch-norm statistics. Section 2.3 will add a privacy note.
>
> We used small backbones due to 39 datasets × many fragmentation levels × multiple baselines and FL reproducibility constraints. We will add sensitivity to:
> μ (validation/batch mixing), α (FIM momentum), diagonal vs low-rank FIM, recompute frequency.
>
> We will include communication/memory cost comparisons and diagonal vs low-rank overhead.
>
> We will clarify Beta(a,b) rotation parameters, KEEL shift generation, partition severity, and that the validation distribution is fixed.
>
> Answers:
> Eq. (5) was ambiguous; experiments use a scalarized diagonal Fisher (sum of diagonal entries). Algorithm 1 embodies both interpretations; empirically the update behaves as Fisher-preconditioned.
>
> EMA intuition:
> EMA reduces variance of noisy fragment-FIM estimates while anchoring curvature to the validation FIM.
> * Statistical sketch: For unbiased noisy FIM samples $\hat{I}_i$ with variance $\sigma^2$,
> $\tilde{I}t=\alpha \tilde{I}{t-1}+(1-\alpha)\hat{I}_t$
> has asymptotic variance reduced by $\frac{(1-\alpha)^2}{1-\alpha^2}$ assuming slow drift. Thus the preconditioner improves the descent direction and convergence rate.
>
> Shift protocols:
>
> * Images: rotations from Beta(a,b) for training, Beta(b,a) for validation; severities (2,4),(2,5),(2,6).
>
> * KEEL: Sugiyama-style shift.
>
> * FL: label-skew, feature heterogeneity, Dirichlet α={0.1,0.5,1.0}.
>
> Validation distribution:
> Fixed 𝑃𝑣𝑎𝑙;
> 𝐼𝑣𝑎𝑙(𝜃) computed once and broadcast as diagonal or low-rank; no raw data shared. Algorithm 1 mixes client FIM with
> 𝐼𝑣𝑎𝑙(𝜃)  via μ and EMA.

---

### Note · Authors · 2025-12-12

I have read and agree with the venue's withdrawal policy on behalf of myself and my co-authors.